# A trans-synaptic IgLON adhesion molecular complex directly contacts and clusters a nicotinic receptor

Morgane Mialon[1], Liubov Patrash [1], Laure Granger[1], Alexis Weinreb[2,3], Engin Özkan [4,5], Jean-Louis Bessereau [1,6] ✉ & Berangere Pinan-Lucarre [1,6] ✉

The clustering of neurotransmitter receptors at appropriate postsynaptic sites is essential for controlling synaptic transmission. While most known mechanisms involve receptor binding with cytoplasmic scaffolds, recent evidence highlights the importance of extracellular interactions that directly target receptors. Using *Caenorhabditis elegans*, we identified a trans-synaptic complex that involves RIG-5 and ZIG-8, two adhesion molecules of the immunoglobulin (Ig) superfamily and orthologous to *Drosophila* DIPs and Dprs, and mammalian IgLONs. Our results show that RIG-5 and ZIG-8 are anchored in the pre- and postsynaptic membranes, respectively, and interact in vivo via their first Ig domains. Furthermore, ZIG-8 directly binds a α7-like acetylcholine receptor (AChR), known as ACR-16, via a *cis*-interaction between its Ig2 domain and the base of the extracellular AChR domain. This study provides direct evidence that trans-synaptic IgLON interactions can organize neurochemical synapses and suggests that the IgLONs may directly interact with ionotropic receptors in the mammalian nervous system.

Chemical neurotransmission relies on the precise anchoring and clustering of postsynaptic receptors directly opposite specific neurotransmitter release sites. This process is tightly controlled by cell surface and secreted molecules that interconnect the molecular components of the synapse and ensure accurate alignment between the presynaptic and postsynaptic domains[1,2]. However, the diversity of molecular players that govern receptor clustering is becoming increasingly complex, raising new questions about the mechanisms regulating synapse formation and homeostasis.

One prevailing mechanism of postsynaptic receptor clustering involves the recruitment of intracellular protein scaffolds, composed of multimodular proteins of the postsynaptic density that multimerize and anchor postsynaptic receptors through intracellular binding. Synaptic transmembrane proteins localize these scaffolds, resulting in

the formation of large molecular assemblies that span the postsynaptic membranes. In the mammalian brain, prototypical examples of these cytoplasmic scaffolds include PSD95 (PostSynaptic Density protein 95), SAP97, Homer and Shank at excitatory synapses, and Gephyrin and Collybistin at inhibitory synapses[3,4].

More recently, an increasing number of components of the synaptic extracellular matrix have been demonstrated to serve as scaffolding molecules and help recruit synaptic receptors through extracellular interactions (reviewed in ref. 5). For example, secreted neuronal pentraxins (Nptx), and the transmembrane-bound neuronal pentraxin receptor (Nptxr) associate with AMPARs via their pentraxin domains in vitro, and cluster AMPARs in vivo, likely through self-multimerization[6,7]. Similarly, Cerebellin 1 (Cbln1), Cerebellin 4 (Cbln4) and the C1ql2-3 complement-related molecules are secreted from

---

[1]Universite Claude Bernard Lyon 1, MeLis, CNRS UMR5284, INSERM U1314, Faculte de Medecine et de Pharmacie, Lyon, France. [2]Department of Genetics, Yale University School of Medicine, New Haven, CT, USA. [3]Department of Neuroscience, Yale University School of Medicine, New Haven, CT, USA. [4]Department of Biochemistry and Molecular Biology, University of Chicago, Chicago, IL, USA. [5]The Neuroscience Institute, University of Chicago, Chicago, IL, USA. [6]These authors contributed equally: Jean-Louis BESSEREAU, Berangere PINAN-LUCARRE. ✉e-mail: jean-louis.bessereau@univ-lyon1.fr; berangere.pinan-lucarre@univ-lyon1.fr

presynaptic terminals and bind the atypical GluD1-2 glutamate receptor delta subunits and the GluK2-4 kainate receptor subunits, respectively[8–10]. Both Cerebellins and C1ql2-3 are part of large complexes that are stabilized through their own oligomerization and bridge pre- and postsynaptic components.

Interestingly, both mechanisms can be used at the same synapse. In the nematode *C. elegans*, the extracellular matrix MADD-4/Punctin is secreted by motoneurons and localizes two different ionotropic acetylcholine receptors (AChRs) at excitatory neuromuscular junctions (NMJs) via two distinct mechanisms. The levamisole-sensitive acetylcholine receptors (L-AChRs) are heteropentameric AChRs that are activated by the nematode-specific cholinergic agonist levamisole[11]. L-AChRs interact with a bona fide extracellular complex consisting of two secreted proteins, LEV-9 and OIG-4, and the ectodomain of a transmembrane protein, LEV-10; all expressed by muscle cells[12–14]. These proteins form L-AChR-containing microclusters that are recruited and stabilized at synapses, most likely through direct interaction with Punctin in the synaptic cleft[15]. The nicotine-sensitive AChRs (N-AChRs) are homopentameric receptors composed of the ACR-16 subunit, the ortholog of the mammalian α7 AChR subunit[16]. Their synaptic localization depends on their interaction with an intracellular scaffold. At the cholinergic NMJ, Punctin activates the netrin receptor UNC-40/DCC and localizes the transmembrane heparan sulfate proteoglycan receptor Syndecan[17–19]. This dual signaling triggers the subsynaptic recruitment of the scaffolding proteins LIN-2/CASK and FRM-3/FARP1-2, which physically bind the large intracellular loop of the ACR-16 receptor and localize it at the NMJ.

In *C. elegans*, acetylcholine is not only used at NMJs but is the major neurotransmitter throughout the nervous system. Neurons express a large diversity of AChRs. In particular, the gene *acr-16* is readily expressed in several neurons but the mechanisms controlling its subcellular localization are unknown. To identify novel mechanisms of synapse organization, we performed an innovative genetic analysis of ACR-16-containing neuron-neuron synapses. Here we describe a critical role of two cell adhesion molecules, namely RIG-5 (neuRonal IG CAM-5) and ZIG-8 (Zwei IG domain protein-8), which are the sole IgLON orthologs in *C. elegans*. IgLONs are Ig-domain rich extracellular proteins that have often been suggested to act as synaptic adhesion molecules across phyla, but their actual synaptic functions have remained largely unexplored. We found that RIG-5 and ZIG-8 are specifically localized to pre- and postsynaptic membranes, respectively, and interact trans-synaptically through their N-terminal Ig1 domains. Most notably, we demonstrated that ZIG-8 recruits AChRs through a direct physical *cis*-interaction involving its Ig2 domain and the base of the AChR ectodomain. This is an unprecedented mechanism controlling the synaptic localization of an AChR through a direct extracellular interaction with synaptic adhesion molecules.

## Results

### The ACR-16 acetylcholine receptor forms neuronal clusters
To visualize ACR-16 in living animals, we used a knock-in allele expressing a Scarlet-tagged version of ACR-16 (Supplementary Fig. 1a)[17]. As previously described, ACR-16-Scarlet appeared at NMJs as characteristic spreading clusters along the ventral and dorsal sides of the animals. In addition, we also visualized large, round and sharply defined clusters in the ventral nerve cord (Supplementary Fig. 1b). These two types of clusters followed distinct tracks. Interestingly, transcriptomic data from single-neuron analyses indicate that *acr-16* is expressed in several neurons within the ventral nerve cord (Supplementary Fig. 1c)[20]. This nerve structure is a major neuropil consisting of a bundle of ~40 neurites and hundreds of en passant synapses that extends from the head to the tail (Fig. 1a)[21]. To investigate whether the round ACR-16 clusters are formed in neurons, we performed tissue-specific degradation of ACR-16 by utilizing the Auxin-Inducible Degradation (AID) system[22]. We introduced the AID cassette in the

tagged *acr-16::Scarlet* locus using CRISPR/Cas9 and triggered proteasomal degradation of ACR-16-AID-Scarlet in body wall muscle cells and neurons (Supplementary Fig. 1a). Degrading ACR-16-AID-Scarlet in body wall muscle cells eliminated the spreading clusters, confirming their presence at NMJs (Fig. 1b). Conversely, degrading ACR-16-AID-Scarlet in neurons caused the disappearance of the round clusters, indicating their neuronal origin. Simultaneous degradation in both neurons and muscle cells completely removed ACR-16-AID-Scarlet, demonstrating highly efficient degradation. For this study, we focused primarily on the neuronal clusters of ACR-16-AID-Scarlet and, unless otherwise specified, we induced ACR-16-AID-Scarlet degradation specifically in body wall muscle cells.

### ACR-16 forms postsynaptic clusters at AVE to AVA synapses
To identify the neurons of the ventral nerve cord in which ACR-16 clusters, we first examined single-neuron transcriptomic data and identified four classes of *acr-16* expressing neurons that potentially synapse in the ventral nerve cord based on neuronal wiring data (Supplementary Fig. 1c)[20,21]. In parallel, we constructed a fluorescent transcriptional reporter driven by a 5.8 kb upstream promoter sequence of *acr-16* (Supplementary Fig. 1a). Its expression pattern was analyzed in the NeuroPAL strain, which facilitates neuron identification through unique color labels and positions[23]. We confirmed *acr-16* expression in two classes of neurons that synapse in the ventral nerve cord: the AVA, a pair of interneurons that control locomotion, and the DB-type of cholinergic motoneurons (Supplementary Fig. 1c, d).

Using fluorescently-tagged ACR-16, we counted on average 116 neuronal clusters along the ventral cord (Fig. 1c, Supplementary Fig. 1e). To specifically evaluate the number of clusters formed in the AVA and DB neurons, we developed a versatile strategy based on split GFP reconstitution in the cytoplasm of specific neurons, which we named NeuroSIL (Neuron-type Specific Illumination, Fig. 1d)[24]. We inserted three copies of the spGFP11 sequence into the *acr-16::aid::Scarlet* locus via CRISPR/Cas9 (*acr-16::aid::Scarlet::spgfp11*) and expressed the complementary spGFP1-10 moiety in the AVA or in the DB neurons (Supplementary Fig. 1a)[25,26]. Using NeuroSIL, the red Scarlet fluorescence labels the full complement of ACR-16 clusters, while the GFP fluorescence is reconstituted only in the neurons expressing spGFP1-10. In the AVA neurons, NeuroSIL revealed on average 108 ACR-16-Scarlet and GFP positive clusters (Fig. 1c, and Supplementary Fig. 1f). Because some clusters showed only Scarlet fluorescence and thus were not formed in the AVAs, we also tested NeuroSIL in the DBs (Fig. 1c, e). This experiment revealed on average 7 GFP+Scarlet clusters in the DB neurons. We concluded that the DBs and AVAs are the two neuron types in which ACR-16 form clusters, with the latter contributing to the majority of these clusters (Fig. 1f).

The AVA and DB neurons receive numerous synaptic inputs from several cholinergic neurons in the ventral nerve cord, as evidenced by electron microscopy (Fig. 1g)[21,27]. Notably, the AVE interneurons establish synapses with the AVA neurons in the first quarter of the ventral nerve cord (Fig. 1f). To investigate the pattern of presynaptic sites in AVE neurons, we built an active zone reporter using CLA-1/Clarinet[28]. Our results revealed the juxtaposition of approximately 16 active zones and ACR-16-AID-Scarlet clusters in the first quarter of the ventral nerve cord, indicating that ACR-16 forms postsynaptic clusters at AVE to AVA synapses (Fig. 1c, g, h). Using the same methodology, we examined a presynaptic AVB reporter, as AVB neurons have been reported to make approximately 27 synapses on AVA by electron microscopy[21,27]. However, we did not detect strong juxtaposition of the pre- and postsynaptic signals in this context, suggesting that ACR-16 is not present at AVB to AVA synapses (Supplementary Fig. 1g). Owing to the lack of specific promoters for certain neuron types, such as AVD, which forms many synapses on AVA, we could not probe every potential presynaptic neuron of the AVAs and DBs[21,27]. Nevertheless, our data identify that ACR-16 form postsynaptic clusters at least at AVE to AVA synapses.

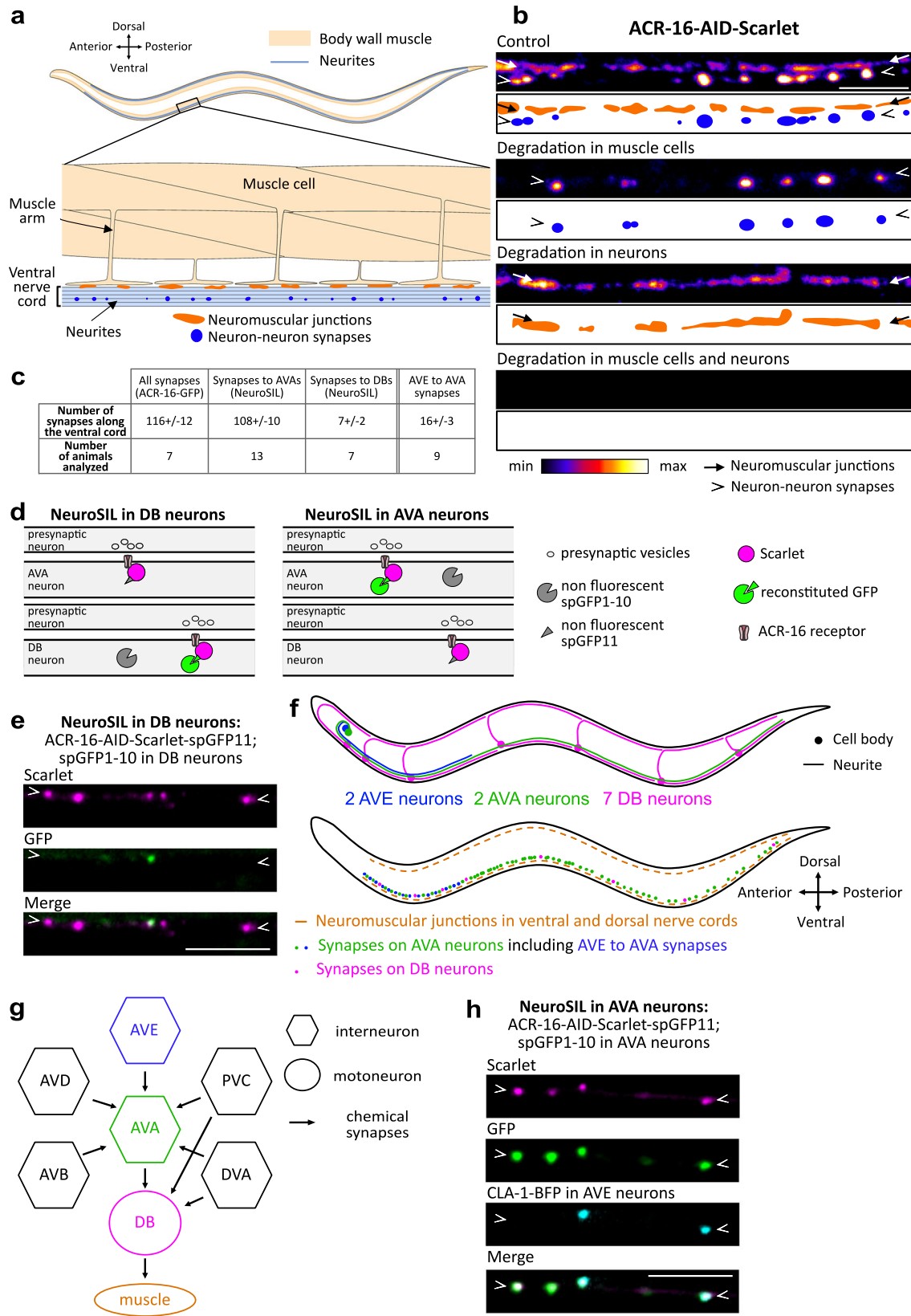

| | All synapses (ACR-16-GFP) | Synapses to AVAs (NeuroSIL) | Synapses to DBs (NeuroSIL) | AVE to AVA synapses |
|---|---|---|---|---|
| **Number of synapses along the ventral cord** | 116+/-12 | 108+/-10 | 7+/-2 | 16+/-3 |
| **Number of animals analyzed** | 7 | 13 | 7 | 9 |

## The RIG-5 and ZIG-8 IgLONs associate with ACR-16

To elucidate the molecular mechanisms controlling ACR-16 synaptic localization, we performed a genetic screen for mutants with altered patterns of ACR-16-AID-Scarlet clustering, and identified several mutations in the *rig-5* and *zig-8* genes (Supplementary Fig. 2a, b). These genes encode orthologs of mammalian IgLON cell adhesion molecules,

also known as DIPs (Dpr Interacting Protein) and Dprs (Defective Proboscis extension Response) in *Drosophila*[29,30]. RIG-5 and ZIG-8 are composed of three and two extracellular Ig-like domains, respectively, such as DIPs and Dprs, and are tethered to the plasma membrane via glycosylphosphatidylinositol (GPI) anchoring[31,32]. Using CRISPR/Cas9, we engineered BFP-RIG-5 and GFP-ZIG-8 translational reporters by

**Fig. 1 | ACR-16 acetylcholine receptors form synaptic clusters in neurons of the ventral nerve cord. a** Schematic of an adult *C. elegans* with a zoom-in showing body wall muscle cells and the ventral nerve cord, including NMJs and neuron-neuron synapses. **b** Spinning disk microscopy images and schematics of ACR-16-AID-Scarlet along the ventral cord at NMJs (orange dots, arrows) and neuron-neuron synapses (blue dots, arrowheads) under control conditions and following auxin-induced degradation in muscle cells, neurons and both. In each figure panel, arrowheads mark the location of neurites expressing ACR-16. **c** Number of ACR-16 clusters along the entire ventral nerve cord (ACR-16-GFP), in AVA and in DB neurons, and at AVE to AVA synapses (colocalizing with a CLA-1-BFP presynaptic marker expressed in AVE neurons). **d** Schematic of the NeuroSIL strategy in AVA and DB neurons in the ventral nerve cord using the ACR-16-AID-Scarlet-spGFP11 strain.

Upon auxin-induced degradation in muscle, all neuronal ACR-16 clusters are marked with Scarlet fluorescence (magenta), while ACR-16 clusters formed in AVAs or in DBs, are also detectable following neuron-specific co-expression of a spGFP1-10 moiety (green). **e** NeuroSIL images in DB neurons. **f** Schematic of an adult *C. elegans* from a lateral view showing the indicated neurons, and synapses with ACR-16. **g** Connectivity diagram of AVA neurons, DB neurons and some of their pre-synaptic partners at chemical synapses in the ventral nerve cord, according to White et al., 1986[21]. **h** NeuroSIL images in AVA neurons showing Scarlet fluorescence and GFP reconstruction, along with presynaptic expression of CLA-1-BFP in AVE neurons. Some graphical elements in Fig. 1d were provided by Servier Medical Art (https://smart.servier.com), licensed under CC BY 4.0 (https://creativecommons.org/licenses/by/4.0/). Scale bars: 5 μm (b, e and h).

tagging them with fluorescent proteins immediately after their signal peptides. Strikingly, both BFP-RIG-5 and GFP-ZIG-8 exhibited round puncta that overlapped extensively with ACR-16-AID-Scarlet clusters along the ventral nerve cord (Fig. 2a). Despite the abundance of cholinergic synapses in the ventral nerve cord, BFP-RIG-5 and GFP-ZIG-8 were only present along with ACR-16-AID-Scarlet, and not at other types of synapses (Fig. 2a, Supplementary Fig. 1h). This evidence strongly supports the conclusion that RIG-5 and ZIG-8 are cell adhesion molecules specifically associated with ACR-16 clusters.

## RIG-5 and ZIG-8 control each other's localization

A peculiar feature of IgLONs is their involvement in multiple homo- and heterophilic interactions between members of the family, in *trans* across different cells[30,33]. For several dimers, the structural basis of these interactions has been solved through crystallography, revealing a central hydrophobic patch of amino acids in the first N-terminal Ig1 domains as crucial for binding[30,34,35]. RIG-5 and ZIG-8 were demonstrated to form heterodimers by interacting in their Ig1 domains, as evidenced by crystallography and binding analysis, akin to mammalian and *Drosophila* IgLONs[31]. Given the in vitro interaction between RIG-5 and ZIG-8, we tested whether this interaction applies in vivo in the synaptic cleft. Firstly, we performed a GFP Reconstruction Across Synaptic Partners (GRASP) experiment by fusing RIG-5 and ZIG-8 to split GFP fragments. The resulting *spgfp11::rig-5* and *spgfp1-10::zig-8* double knock-in strain exhibited robust GFP fluorescence that precisely overlapped with ACR-16-AID-Scarlet neuronal puncta, thereby indicating the proximity of RIG-5 and ZIG-8 in the synaptic cleft (Fig. 2b).

Secondly, we assessed the impact of *rig-5* deletion on GFP-ZIG-8 localization. To completely abolish the *rig-5* gene activity, we engineered a full deletion mutant using CRISPR/Cas9 (Supplementary Fig. 2a). In this *rig-5(0)* mutant, GFP-ZIG-8 failed to concentrate at synapses in the ventral nerve cord, instead exhibiting a diffuse distribution along the neurites (Fig. 2c). Similarly, analyzing the distribution of a GFP-RIG-5 reporter in a *zig-8(0)* deletion allele revealed a dramatic loss of synaptic concentration (Supplementary Fig. 2b, Fig. 2d). This underscores the essential roles of both RIG-5 and ZIG-8 in maintaining their synaptic localization.

Thirdly, we introduced point mutations in residues within the Ig1 domains previously demonstrated to disrupt the interaction between RIG-5 and ZIG-8[31]. We engineered a *gfp::rig-5(F75D)* allele and observed that both GFP-RIG-5(F75D) and Scarlet-ZIG-8 exhibited diffuse distributions along the neurites (Fig. 2e)[31]. This suggested that disrupting the interaction between RIG-5 and ZIG-8 affects their synaptic localization. In a reciprocal experiment, we engineered a *Scarlet::zig-8(L77E)* allele, which also resulted in a diffuse distribution of both Scarlet-ZIG-8(L77E) and GFP-RIG-5 along neurites (Fig. 2e). Likewise, GFP-RIG-5(F75D) and Scarlet-ZIG-8(L77E) failed to concentrate at synapses in a *gfp::rig-5(F75D); Scarlet::zig-8(L77E)* double mutant (Fig. 2e). Taken together, these experiments strongly support the notion that the interaction between RIG-5 and ZIG-8 in the synaptic cleft is crucial for their synaptic localization.

## RIG-5 and ZIG-8 control ACR-16 clustering

Having identified RIG-5 and ZIG-8 in a genetic screen for mutants in ACR-16 localization, we aimed to precisely characterize their synaptic function (Supplementary Fig. 2a). In the *rig-5(0)* and *zig-8(0)* null mutants, as well as in the double *rig-5(0); zig-8(0)* mutant, ACR-16-AID-Scarlet failed to cluster at neuron-neuron synapses (Fig. 3a). In contrast, the clustering of an ACR-16-AID-Scarlet reporter occurred normally at NMJ in *rig-5(0)* and *zig-8(0)* mutants (Supplementary Fig. 3a). This result is consistent with the known clustering mechanism of ACR-16 in muscle cells, which involves an intracellular scaffold composed of FRM-3/FARP and LIN-2/CASK[17]. To verify that the neuronal phenotype was independent of the Scarlet tag, we analyzed the *rig-5(0)* and *zig-8(0)* mutations using an ACR-16-AID-GFP knock-in reporter, and found similar clustering defects at neuronal synapses (Supplementary Fig. 3b). Additionally, given that the RIG-5(F75D) and ZIG-8(L77E) variants lose the ability to interact in vivo and accumulate at synapses, we investigated the impact of these mutations on ACR-16-AID-Scarlet distribution. As expected, ACR-16-AID-Scarlet did not cluster at synapses in these missense mutants (Fig. 3b, c). We concluded that RIG-5 and ZIG-8 control the concentration of ACR-16 specifically at neuron-neuron synapses of the ventral nerve cord, but not at NMJs.

Conversely, we investigated whether RIG-5 and ZIG-8 could concentrate at synapses in the absence of *acr-16*. We found that GFP-RIG-5 and GFP-ZIG-8 formed dimmer puncta at synapses, but also displayed a diffuse pattern between synapses in *acr-16* mutants (Fig. 3d, e). This suggests that ACR-16 is not critical for recruiting RIG-5 and ZIG-8 to synapses but may play a role in stabilizing their complex.

Synaptic adhesion molecules function during development and at mature synapses. To investigate whether RIG-5 and ZIG-8 have a developmental role, we analyzed larval stage 1 (L1) *C. elegans*. In newly born L1 larvae, BFP-RIG-5, GFP-ZIG-8 and ACR-16-AID-Scarlet highly colocalized in a punctate pattern, as in adult *C. elegans* (Fig. 2a, Supplementary Fig. 4a). Moreover, we observed that *rig-5(0)* and *zig-8(0)* mutations prevented ACR-16-AID-Scarlet clustering (Supplementary Fig. 4b). Additionally, GFP-RIG-5 exhibited a diffuse pattern in the *zig-8(0)* mutant (Supplementary Fig. 4c). Similarly, GFP-ZIG-8 showed a diffuse pattern in the *rig-5(0)* mutant (Supplementary Fig. 4d). The consistent findings observed during both larval and adult stages suggest that RIG-5 and ZIG-8 control ACR-16 clustering throughout synapse development and maintenance.

The development of neuronal networks involves multiple coordinated steps, including neuronal specification, migration, axon growth and guidance, synaptic partner recognition, and synapse formation and maintenance. To explore the potential involvement of RIG-5 and ZIG-8 in the initial phases of neuronal network establishment, we assessed *rig-5(0)* and *zig-8(0)* single mutants for defects in neuronal specification, guidance and maintenance of the AVE, AVA and DB neurons. Our results revealed that these mutants did not exhibit noticeable defects in these processes (Supplementary Fig. 5a–e). These results are consistent with previous studies showing that RIG-5 and ZIG-8 contribute to nervous system development and maintenance,

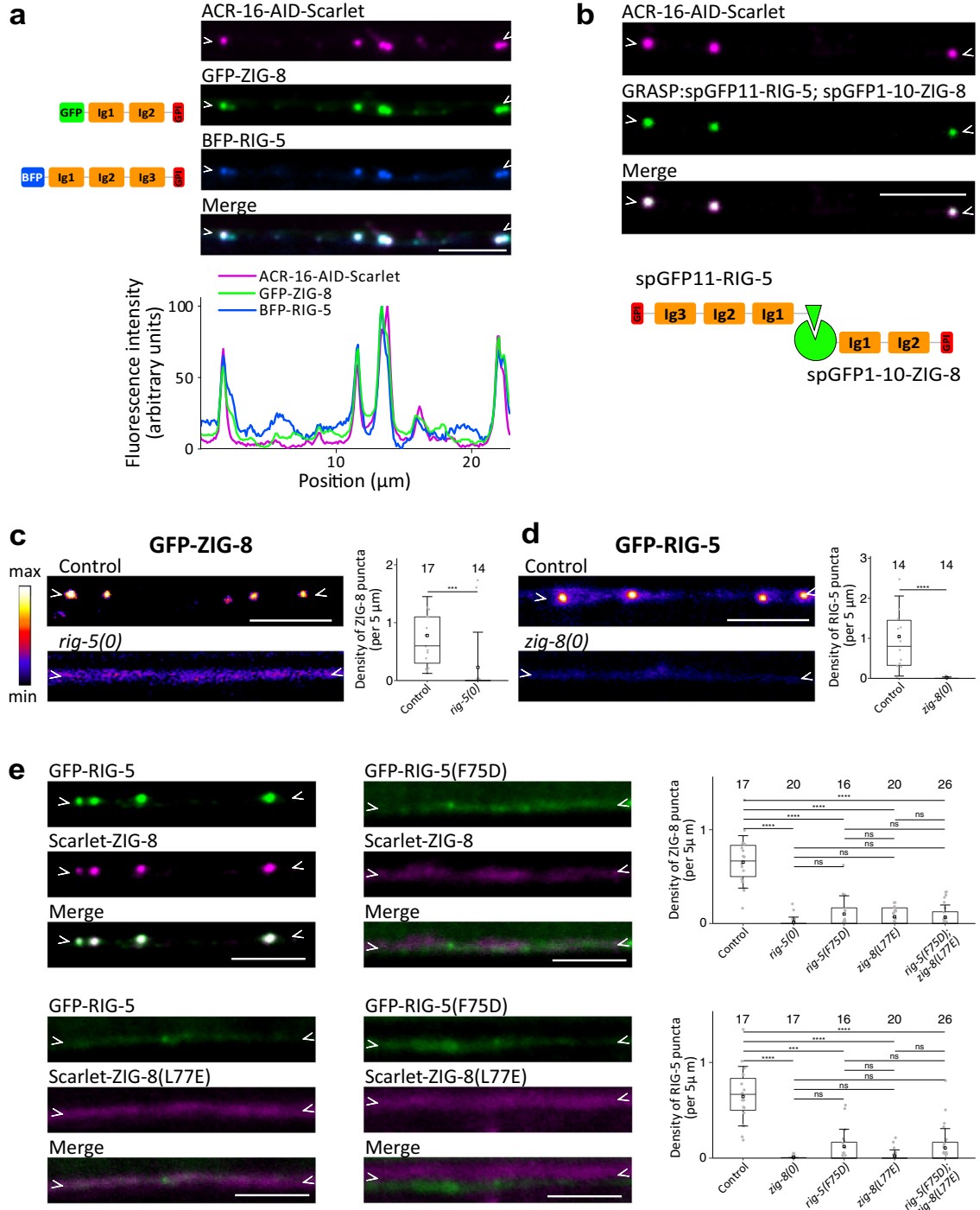

**Fig. 2 | RIG-5 and ZIG-8 interact at synapses that express ACR-16. a** Knock-in reporters of BFP-RIG-5 and GFP-ZIG-8 show strong colocalization with neuronal ACR-16-AID-Scarlet upon auxin-induced degradation in muscle. The fluorescence intensity profile is shown below. Left: schematic of BFP-RIG-5 and GFP-ZIG-8 functional domains. **b** Neuronal ACR-16-AID-Scarlet clusters observed following auxin-mediated degradation of ACR-16 in muscle, and GRASP between RIG-5 and ZIG-8 as indicated. **c, d** Fluorescence images and puncta density of GFP-ZIG-8 in a control strain and in the *rig-5(0)* mutant (**c**), and of GFP-RIG-5 in a control strain and in the *zig-8(0)* mutant (**d**). **e** GFP-RIG-5 and Scarlet-ZIG-8 were observed in control condition, and using strains bearing missense mutations that prevent the RIG-5– ZIG-8 interaction: *gfp::rig-5(F75D)* and *Scarlet::zig-8(L77E)* single mutants, and the *gfp::rig-5(F75D); Scarlet::zig-8(L77E)* double mutant. The density of RIG-5 and ZIG-8 puncta is shown. Data are presented as boxplots showing lower and upper quartiles (box), mean (square), median (center line) and standard deviation (whiskers); the number of worms is indicated for each condition; two-sided Mann-Whitney U test (**c**, **d**), Kruskal-Wallis and Dunn's test (**e**); ns: non-significant, ***< 0.0005, ****< 0.00005. For this and subsequent figures, the *p* values from the statistical tests are provided in the Source Data file. Scale bars: 5 μm. Arrowheads delineate neurites of the ventral nerve cord where synapses may form.

but in strict redundancy with other cell surface receptors and extra-cellular matrix components[36–38]. Taken together, our data suggest that RIG-5 and ZIG-8 control the clustering of ACR-16 nicotinic receptors at specific cholinergic synapses.

## RIG-5 is presynaptic, and ZIG-8 postsynaptic
Our analysis has identified ACR-16 clusters in the AVA and DB neurons, notably at AVE to AVA synapses. This allowed us to investigate in which neurons *rig-5* and *zig-8* function. Using a

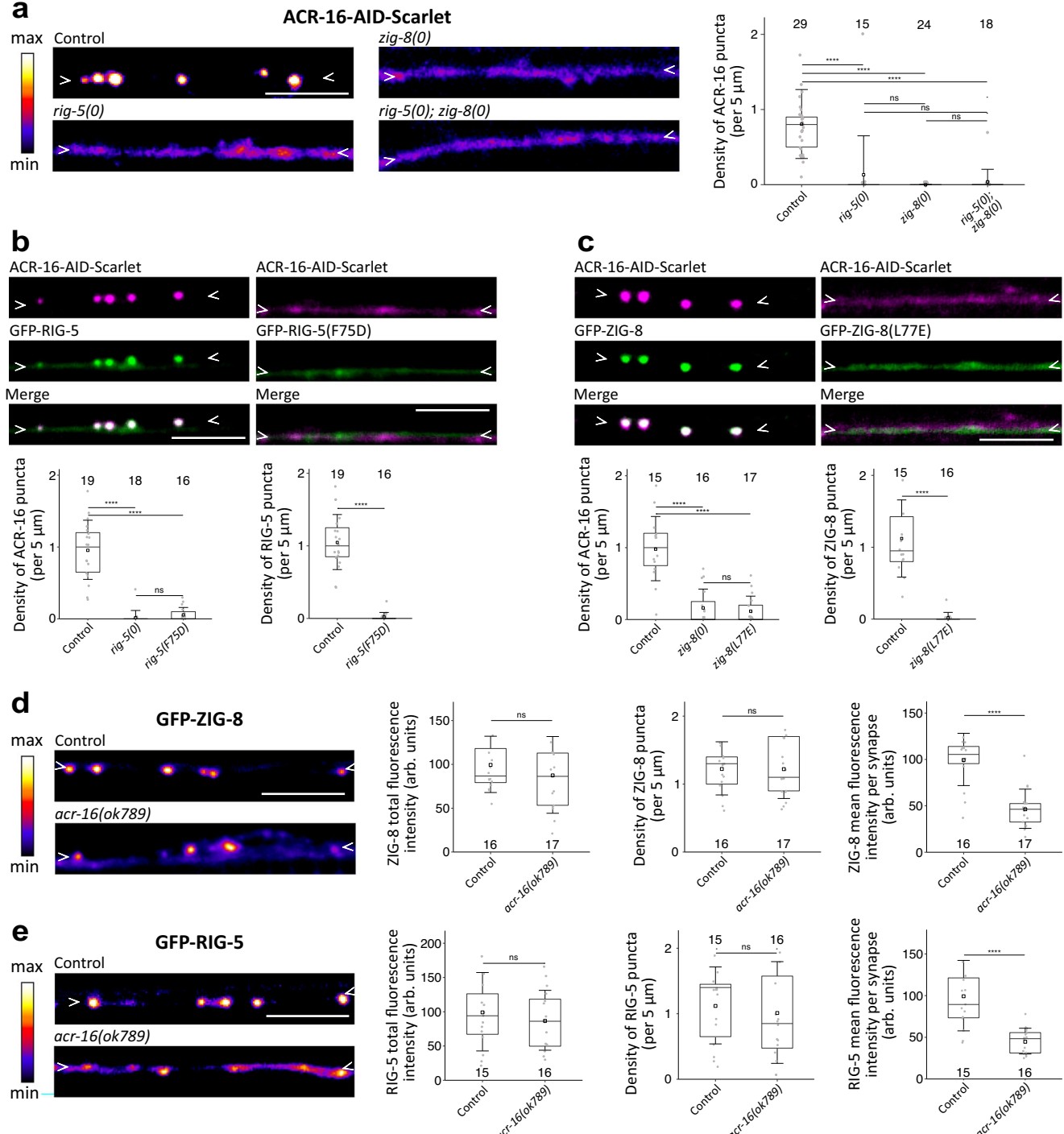

**Fig. 3 | RIG-5 and ZIG-8 control ACR-16 clustering. a** ACR-16-AID-Scarlet fluorescence shows clustering defects in neurons of the *rig-5(0)* and *zig-8(0)* mutants, as well as in the *rig-5(0); zig-8(0)* double mutant. The density of ACR-16 puncta is shown. **b** Clusters of ACR-16-AID-Scarlet and GFP-RIG-5 do not form when a mutation that prevents the RIG-5–ZIG-8 interaction is introduced (*gfp::rig-5(F75D)*). The density of ACR-16 and RIG-5 puncta is shown. **c** Clusters of ACR-16-AID-Scarlet and GFP-ZIG-8 do not form in the *gfp::zig-8(L77E)* mutant. The density of ACR-16 and ZIG-8 puncta is shown. **d**, **e** The GFP-ZIG-8 (**d**) and GFP-RIG-5 (**e**) fluorescence patterns were assessed in control and *acr-16(ok789)* mutant backgrounds. The total

fluorescence intensity, puncta density and fluorescence intensity per synapse are shown. In this figure, an auxin treatment was applied to degrade ACR-16 in muscle cells. Data are presented as boxplots showing lower and upper quartiles (box), mean (square), median (center line) and standard deviation (whiskers); the number of worms is indicated for each condition; Kruskal-Wallis and Dunn's test (a, b.1, c.1), two-sided Mann-Whitney U test (b.2, c.2, d.1, e.3), two-sided Student's t-test (d.2,3, e.1,2); ns: non-significant, ****< 0.00005. Scale bars: 5 µm. Arrowheads delineate neurites of the ventral nerve cord where synapses may form.

neuronal gene expression atlas, we found that either *rig-5*, *zig-8*, or both are expressed in AVAs, DBs and most of their known synaptic partners (Supplementary Fig. 1c)[20]. Specifically, *rig-5* was expressed in the AVE, AVD, DVA and AVA neurons, while *zig-8* was expressed

in the AVE, AVA and DB neurons. The DB neurons exclusively express *zig-8*, suggesting a model in which RIG-5 might be presynaptic, and ZIG-8 might be postsynaptic at synapses onto AVAs and DBs.

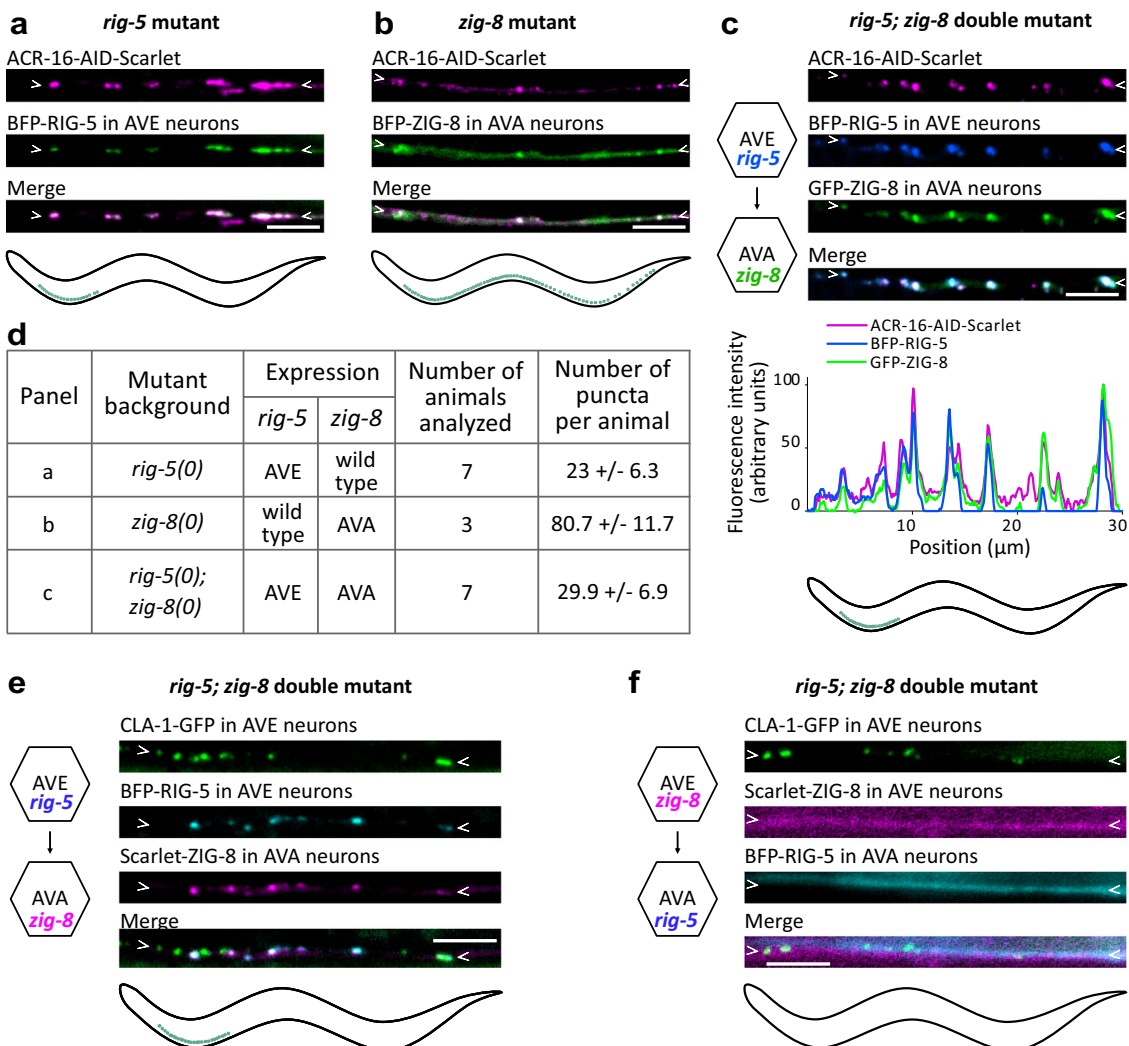

**Fig. 4 | RIG-5 functions in presynaptic neurons and ZIG-8 in postsynaptic neurons. a** Rescue of ACR-16-AID-Scarlet clustering in *rig-5(0)* mutants by expressing BFP-RIG-5 in AVE neurons. **b** Rescue of ACR-16-AID-Scarlet clustering in *zig-8(0)* mutants by expressing BFP-ZIG-8 in AVA neurons. **c** Rescue of ACR-16-AID-Scarlet clustering in *rig-5(0); zig-8(0)* double mutants by expressing both BFP-RIG-5 in AVE neurons and GFP-ZIG-8 in AVA neurons. The fluorescence intensity profile is shown below. Left: diagram representing cell-specific expression. **d** Number of ACR-16-AID-Scarlet clusters observed in the rescue strains (shown in **a**–**c**) along the ventral nerve cord. Data are shown as mean +/-SD. **e** Clusters of BFP-RIG-5 and Scarlet-ZIG-8 face an AVE-specific active zone marker (CLA-1-GFP), in *rig-5(0); zig-8(0)* double mutants expressing both BFP-RIG-5 in AVE neurons and Scarlet-ZIG-8 in AVA neurons. **f** In *rig-5(0); zig-8(0)* double mutants expressing both BFP-RIG-5 in AVA neurons and Scarlet-ZIG-8 in AVE neurons, no clusters are formed despite the presence of the AVE-specific CLA-1-GFP reporter. In panels a-c, an auxin treatment was applied to degrade ACR-16 in muscle cells. Scale bars: 5 μm. Arrowheads delineate neurites of the ventral nerve cord where synapses may form.

To test this hypothesis, we performed a series of cell-specific rescue experiments, in single *rig-5(0)*, *zig-8(0)* or double *rig-5(0); zig-8(0)* mutant backgrounds, in which ACR-16 does not cluster properly (Fig. 3a). In these rescue experiments, we observed the recovery of distinct classes of synapses depending on the genetic background. Firstly, we expressed BFP-RIG-5 in the AVE neurons of a *rig-5(0)* mutant and found that BFP-RIG-5 had a punctate distribution typical of AVE to AVA synapses, forming more than 20 puncta in the first quarter of the ventral nerve cord (Fig. 4a, d, Fig. 1c, f). These BFP-RIG-5 puncta robustly colocalized with ACR-16-AID-Scarlet, suggesting that presynaptic expression of RIG-5 could rescue postsynaptic ACR-16 localization. Secondly, we introduced BFP-ZIG-8 in the AVAs of a *zig-8(0)* mutant. We observed an average of 80 puncta containing both BFP-ZIG-8 and ACR-16-AID-Scarlet along the ventral nerve cord. They likely correspond to synapses where ACR-16 clusters in AVA neurons, from various presynaptic neurons, including but not limited to AVE neurons (Fig. 4b, d, Fig. 1c, f). This experiment suggested that expression of ZIG-8 in the postsynaptic AVA neuron could promote ACR-16-AID-Scarlet

clustering. Thirdly, we co-expressed BFP-RIG-5 in the AVE neurons and GFP-ZIG-8 in the AVA neurons in the double *rig-5(0); zig-8(0)* mutant. We observed a sharp punctate distribution of BFP-RIG-5, GFP-ZIG-8, and ACR-16-AID-Scarlet, consistent with the synaptic pattern observed at AVE to AVA connections (Fig. 4c, d, Fig. 1c, f).

To confirm the synaptic location of these puncta, we expressed BFP-RIG-5 in the AVEs along with an AVE-specific presynaptic marker, and Scarlet-ZIG-8 in the AVA, in the double *rig-5(0); zig-8(0)* mutant. We observed that BFP-RIG-5 and Scarlet-ZIG-8 formed puncta juxtaposed to the AVE presynaptic marker, supporting the formation of functional synapses (Fig. 4e). To probe whether the orientation of the RIG-5−ZIG-8 trans-synaptic complex is important, we reversed their expression, introducing Scarlet-ZIG-8 in the presynaptic AVE neuron, and BFP-RIG-5 in the postsynaptic AVA neuron. Strikingly, we did not observe puncta of BFP-RIG-5 or Scarlet-ZIG-8 in this genetic background (Fig. 4f). This finding underscores that RIG-5 must be expressed in the presynaptic neuron and ZIG-8 in the postsynaptic neuron for correct localization to their respective pre- or postsynaptic membranes and

efficient interaction at synapses. Overall, this set of cell-specific rescue experiments indicates that RIG-5 is presynaptic, and ZIG-8 post-synaptic, at least at AVE to AVA synapses.

Notably, the number of puncta observed in rescue experiments involving transgenic overexpression is comparable, though not identical, to those quantified using endogenous knock-in reporters. For instance, in the *rig-5; zig-8* double mutant background, expression of BFP-RIG-5 in AVE and Scarlet-ZIG-8 in AVA neurons resulted in 29.9 ± 6.9 AVE-to-AVA puncta (Fig. 4c, d), whereas under physiological conditions, only 16 ± 3 AVE-to-AVA synapses were observed (Fig. 1c, h). Such increase is likely caused by transgenic overexpression, suggesting that the precise temporal and quantitative regulation of RIG-5 and ZIG-8 is essential for establishing the correct number of synapses.

### The Ig2 domain of ZIG-8 recruits ACR-16

RIG-5 and ZIG-8 are adhesion proteins composed of 3 and 2 Ig-like domains, respectively. We analyzed the function of truncated RIG-5 and ZIG-8 proteins generated by CRISPR/Cas9 gene genome engineering, except for deletions of their Ig1 domains, which prevented their expression and were therefore not studied. Since ZIG-8 is required at the postsynaptic membrane, we investigated the role of its Ig2 domain in ACR-16-AID-Scarlet localization. Strikingly, in the absence of its Ig2 domain (*gfp::zig-8(ΔIg2)*), ZIG-8ΔIg2 formed synaptic clusters that overlapped with BFP-RIG-5, yet ACR-16-AID-Scarlet remained diffuse along the neurites (Fig. 5a). To exclude that this could be caused by the shortening of the protein, we replaced the ZIG-8 Ig2 domain with the human CD4 Ig2 domain (*gfp::zig-8(ΔIg2)::CD4(Ig2)*) and observed that ACR-16-AID-Scarlet also failed to cluster properly (Fig. 5a). These results indicate that the Ig2 domain of ZIG-8 is critical for recruiting ACR-16. In contrast, fragments of RIG-5 showing deletions in the Ig2, Ig3, or both Ig2 and Ig3 domains, were detected at synapses and effectively promoted ACR-16-AID-Scarlet concentration in the presence of full-length ZIG-8 (Supplementary Fig. 2 and 6a–c).

Our results were consistent with a direct interaction between ZIG-8 and ACR-16. To test this hypothesis, we ran a structure prediction analysis using AlphaFold-Multimer. ACR-16 adopted a typical AChR structure, forming a pentamer composed of five subunits arranged around a central pore (Fig. 5b and Supplementary Fig. 7a)[39]. Interestingly, this analysis indicated potential interactions between ZIG-8 and each individual ACR-16 subunit at the base of the large N-terminal extracellular domain. The predicted interface involves several unconventional side-to-side, strand-to-strand hydrogen bond interactions, with the beta-strand G of ZIG-8 interacting with the beta-strand 9 of ACR-16 (Fig. 5c). This coupling is likely further stabilized by additional hydrogen bond and hydrophobic interactions involving atoms in the main and side chains of surrounding residues. For example, we observed that T201 and L199 side chains of ACR-16 sit in a mostly hydrophobic cavity formed between the A and G strands of ZIG-8's Ig2 domain (Fig. 5d). Based on this, we predict that a bulkier side chain at these positions, such as an arginine at T201, would break this interaction.

To validate this predicted ZIG-8–ACR-16 interaction interface, we introduced a T201R point mutation into *acr-16::aid::Scarlet* (*acr-16(T201R)::aid::Scarlet*). To test whether ACR-16(T201R) was still functional, we used a loss-of-function mutation in *unc-29*, which encodes an essential subunit of the heteromeric L-AChRs present at the NMJ. When homomeric and heteromeric AChRs were non-functional in an *unc-29; acr-16* double mutant, animals were paralyzed, whereas the *unc-29* single mutant exhibited mild locomotion defects and the *acr-16* single mutant displayed normal locomotion (Supplementary Fig. 8). Importantly, the *acr-16(T201R)* mutant showed normal locomotion and did not exacerbate locomotion defects when combined with the *unc-29* mutation. These results indicated that the *acr-16(T201R)* mutation did not impair ACR-16 expression and function. To probe the interaction of the ACR-16(T201R)-AID-Scarlet variant with ZIG-8, we examined its clustering at neuron-neuron synapses in vivo. We found that the ACR-16(T201R)-AID-Scarlet variant failed to properly cluster at synapses labeled with BFP-RIG-5 and GFP-ZIG-8 (Fig. 5e). Our data strongly suggest that the T201R mutation in *acr-16* disrupts its ability to interact with ZIG-8 in vivo at synapses.

To investigate whether ZIG-8 can directly cluster ACR-16 in vivo, we took advantage of the fact that ACR-16 is also expressed in muscle cells, but does not rely on RIG-5 and ZIG-8 for its synaptic localization. Rather, it involves an interaction of the large intracellular loop of ACR-16 with an intracellular scaffold comprising the FRM-3 multimodular protein. In *frm-3* null mutants, ACR-16-Scarlet is still present at the plasma membrane of muscle cells but fails to concentrate at NMJs and is not detected by fluorescence microscopy (Fig. 6b)[17]. To test whether ZIG-8 is sufficient to cluster ACR-16, we engineered a mNeon-Green(mNG)-tagged chimeric protein between the ectodomain of the adhesion molecule NLG-1/neuroligin, which we previously demonstrated to localize to GABAergic NMJs of *C. elegans*, and the Ig2 domain of ZIG-8 with its GPI anchor signal (Fig. 6a, c)[18]. Strikingly, expression of the mNG-NLG-1-ZIG-8(Ig2) chimera in the muscle cells of *frm-3* mutants caused the clustering of ACR-16-Scarlet at GABAergic NMJs. To confirm that the formation of these clusters did involve the ZIG-8–ACR-16 extracellular interface, we performed two control experiments. First, we introduced the T201R mutation in ACR-16-Scarlet and observed that mNG-NLG-1-ZIG-8(Ig2) no longer recruited ACR-16 at GABAergic NMJs (Fig. 6d, f–h). Second, we replaced the Ig2 domain of ZIG-8 by its Ig1 domain in the chimera and, although the mNG-NLG-1-ZIG-8(Ig1) chimera still localized to GABAergic NMJs, it failed to cluster ACR-16 (Fig. 6e, f–h).

Taken together, these results support a direct mechanism whereby the Ig2 domain of ZIG-8 binds the base of the ACR-16 extracellular domain *in cis* on the postsynaptic membrane to cluster ACR-16 at neuron-neuron synapses.

## Discussion

By focusing on cholinergic neuronal synapses in *C. elegans*, we have uncovered an original paradigm for synaptic organization. Our unbiased genetic screen identified mutations in the *rig-5* and *zig-8* genes that cause dramatic defects in ACR-16 clustering. Our data indicate that RIG-5 and ZIG-8, the orthologs of the IgLONs cell adhesion molecules, form a trans-synaptic bridge within the synaptic cleft in vivo via their Ig1 domains. RIG-5 is tethered to the presynaptic membrane, whereas ZIG-8 is bound to the postsynaptic membrane. Remarkably, the Ig2 domain of ZIG-8 directly interacts with the extracellular domain of ACR-16 to localize this AChR at synapses. Altogether, our results uncover a mechanism of molecular assembly wherein a pair of IgLON cell adhesion molecules interact across synaptic membranes to directly cluster a postsynaptic cholinergic receptor.

Acetylcholine is the most widely used neurotransmitter in *C. elegans*. Accordingly, the acetylcholine-gated ionotropic receptor family has been expanded and potentially contains around 30 AChR subunits that can assemble into homo- and heteromeric receptors[40]. By comparison, the human genome encodes only 16 AChR subunits, despite the many orders of magnitude greater complexity of the human brain. Only a few subunits have been assigned to specific synapses and characterized in vivo in *C. elegans*. The clustering of ACR-16 at the NMJ has been extensively studied. We have previously shown that it relies on an interaction with a synaptic intracellular scaffold comprising FRM-3, a FERM domain-containing protein orthologous to mammalian FARP1/2, and LIN-2, a MAGUK orthologous to CASK[17,19]. This intracellular scaffolding mechanism is strikingly different from the direct extracellular interaction observed with the ZIG-8 Ig2 domain at neuron-neuron synapses. Thus, our results demonstrate that distinct mechanisms have evolved within the same species to control the clustering of a single ionotropic receptor type at different synapses.

In mammals, ionotropic nicotinic AChRs are expressed in a wide variety of neuronal and non-neuronal cell types, regulating various

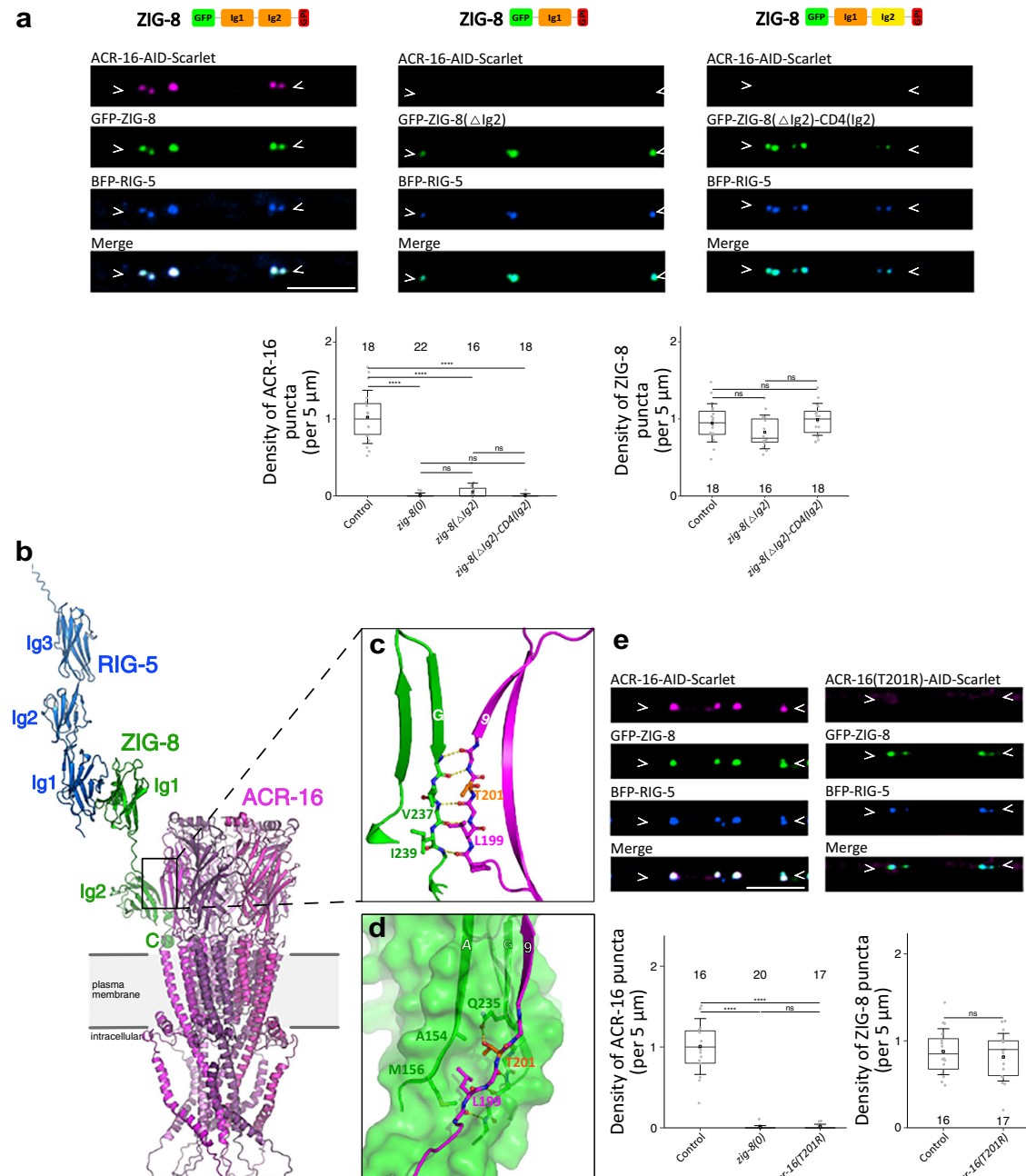

**Fig. 5 | A direct interaction between the ZIG-8 Ig2 domain and the ACR-16 receptor. a** ACR-16-AID-Scarlet receptors cannot form clusters when the Ig2 domain of ZIG-8 is deleted (*gfp::zig-8(ΔIg2)*) or when it is swapped with the Ig2 domain of human CD4 (*gfp::zig-8(ΔIg2)::CD4(Ig2)*). A schematic of ZIG-8 functional domains is shown for each knock-in condition. The density of ACR-16 and ZIG-8 puncta is shown. **b** Cartoon representation of the composite RIG-5−ACR-16−ZIG-8 complex model as predicted by AlphaFold-multimer runs. ACR-16 protomers are drawn in various shades of magenta, while ZIG-8 is green and RIG-5 is blue. The C-terminal putative GPI anchor site of ZIG-8, labelled with a green sphere and letter "C", is placed near the expected plasma membrane by Alphafold. **c** ACR-16 strand 9 and the G strand of ZIG-8 Ig2 domain make an anti-parallel beta sheet. Dashed yellow lines represent H-bonds. **d** Side chains of L199 and T201 in ACR-16 strand 9

sit in a complementary cavity on the ZIG-8 surface created by the A and G chains. Position of the T201 side chain is further stabilized by a hydrogen bond to the side chain of ZIG-8 Q235 (G strand). **e** The T201R mutation in ACR-16 (*acr-16(T201R)::aid::Scarlet*) prevents its clustering, while BFP-RIG-5 and GFP-ZIG-8 formed wild-type puncta. The density of ACR-16 and ZIG-8 puncta was assessed. In this figure, an auxin treatment was applied to degrade ACR-16 in muscle cells. Data are presented as boxplots showing lower and upper quartiles (box), mean (square), median (center line) and standard deviation (whiskers); the number of worms is indicated for each condition; Kruskal-Wallis and Dunn's test (a.1, e.1), one-way ANOVA and Tukey-hsd test (a.2, e.2); *ns* non-significant, ****< 0.00005. Scale bars: 5 μm. Arrowheads delineate neurites of the ventral nerve cord where synapses may form.

biological functions across tissues and systems[41]. In the central nervous system, they play important roles in controlling neural functions, including neuronal network activity and cognitive processing[42]. AChR are mainly present at presynaptic sites in axonal terminals, where they generally increase the release of many different neurotransmitters[43,44].

They are also found in postsynaptic regions and at extrasynaptic sites, where they are likely activated by volume transmission.

The mechanisms underlying the formation and positioning of neuronal AChRs within specific membrane microdomains remain unclear. Some studies have suggested that AChRs can be selectively

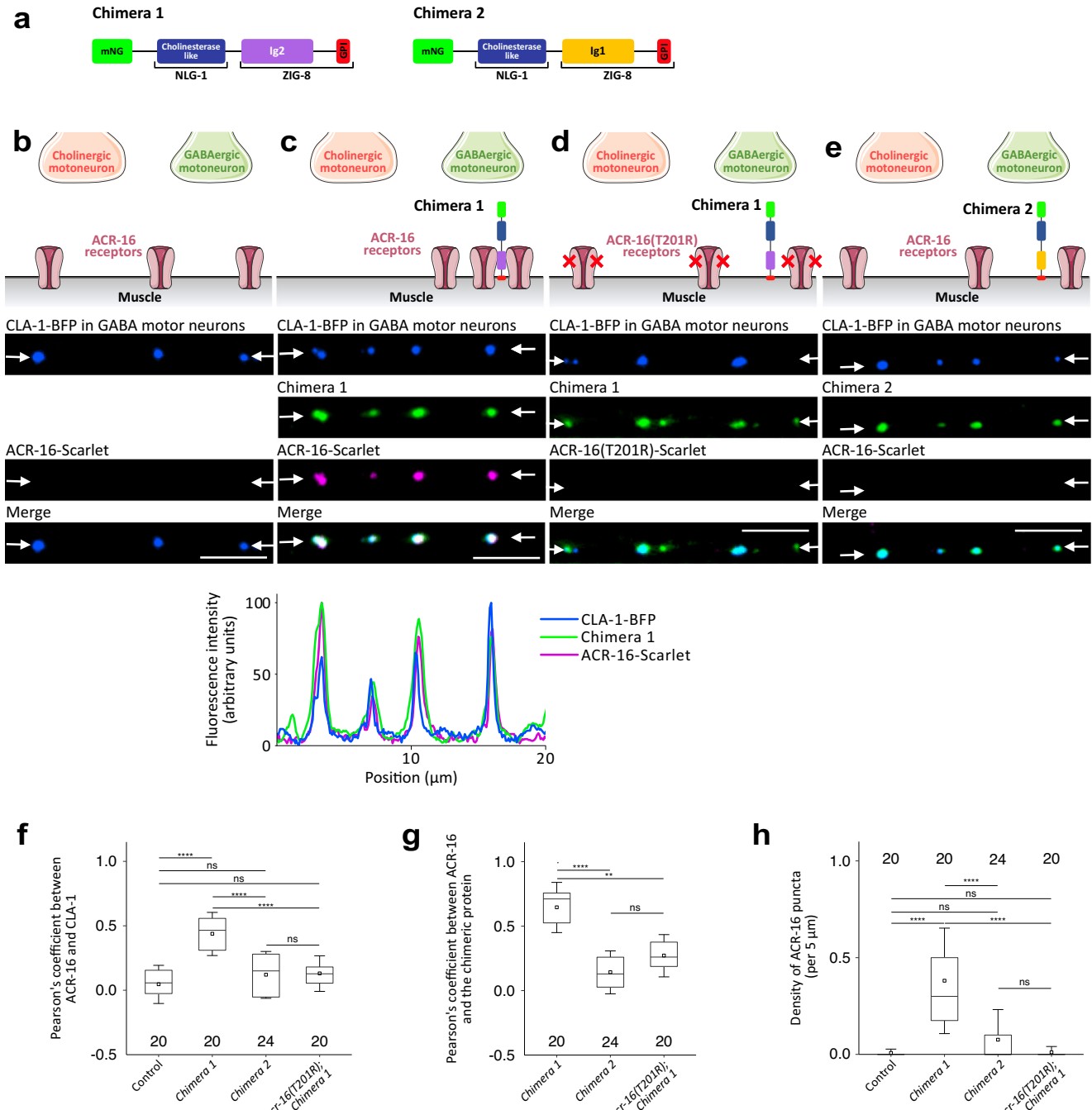

**Fig. 6 | The ZIG-8 Ig2 domain is sufficient for ectopic clustering of the ACR-16 receptor at GABAergic neuromuscular junctions. a** Domain composition of the two chimeric proteins: mNeonGreen (mNG), cholinesterase-like domain of NLG-1/ neuroligin, Ig2 or Ig1 domain, and GPI anchor of ZIG-8. **b** ACR-16-Scarlet fails to cluster at NMJs in the *frm-3(0)* mutant, in the dorsal nerve cord. BFP-CLA-1 is expressed in GABA motor neurons to label GABAergic NMJs. **c** Chimera 1 coloca-lizes with ACR-16-Scarlet at GABAergic NMJs. The fluorescence intensity profile is shown below. **d** Chimera 1 localizes to GABAergic NMJs without ACR-16(T201R)- Scarlet. **e** Chimera 2 localizes to GABAergic NMJs without ACR-16-Scarlet. **f, g** Pearson's coefficients indicate the degree of colocalization (ranging from -1:

anti-correlation, to 1: complete colocalization) between ACR-16 and the BFP-CLA-1 presynaptic reporter (**f**) and between ACR-16 and the chimeric proteins (**g**). **h** The density of ACR-16 puncta is shown. Data are presented as boxplots showing lower and upper quartiles (box), mean (square), median (center line) and standard deviation (whiskers); the number of worms is indicated for each condition; Kruskal-Wallis and Dunn's test (**f**–**h**); *ns* non-significant, **< 0.005, ****< 0.00005. Some graphical elements in Fig. 6b were provided by Servier Medical Art (https://smart. servier.com), licensed under CC BY 4.0 (https://creativecommons.org/licenses/by/ 4.0/). Scale bars: 5 μm. Arrows delineate regions of the dorsal nerve cord where neuromuscular junctions form.

clustered by classical postsynaptic intracellular scaffolds, such as those of the PSD95 family. However, it is still unknown whether this association is direct or indirect[45–47]. In addition, the PICK1 scaffolding protein has been shown to interact with the α7 cytoplasmic loop through an unconventional PDZ recognition mode, and to inhibit α7 surface expression in hippocampal GABAergic neuronal

cultures[48,49]. Finally, PDZ and LIM domain 5 (PDLIM5) binds α7 AChRs via its PDZ domain, increasing their synaptic concentration and enhancing cholinergic neurotransmission[50].

In contrast, no extracellular scaffolds have been described for neuronal AChRs. Nevertheless, AChRs interact with the lynx proto-toxins, non-venomous proteins evolutionarily related to snake

neurotoxins that can bind to the extracellular N-terminal domain of AChRs[51]. These proteins share a distinct "three finger" toxin fold and can be membrane-bound via GPI anchoring or secreted[52]. Due to their distinct binding specificity to AChRs and unique expression patterns in the mammalian brain, immune system, and other tissues, Lynx prototoxins have been proposed to modulate the function of specific AChR subtypes in a regionally specific manner. However, they have not been shown to control AChR localization. In *C. elegans*, it might be interesting in the future to test whether ZIG-8 binding modulates the function of ACR-16 in a similar manner, although ZIG-8 does not interfere with the ACR-16 agonist binding site based on in silico prediction.

Importantly we have identified that AChRs can be clustered within the synaptic cleft by interactions with their extracellular domain. Similar interactions of AChRs with IgLONs or other Ig-domain-containing proteins may control their subcellular localization in mammals.

*rig-5* and *zig-8* are the sole members of the IgLON family in *C. elegans*. Up to now, no phenotype had been described in single null mutants, but discrete phenotypes have been reported for the maintenance of specific axons, in combination with mutants affecting cell surface and extracellular matrix proteins[36–38]. In contrast, this family, known as DIPs and Dprs, has expanded in *Drosophila melanogaster*[29,32]. There are 9 DIP and 21 Dpr paralogs able to form selective homo- and heterodimers through their Ig1 domains with varying affinities, such as in mammalian IgLONs[29,53–55]. Both DIPs and Dprs are present throughout the *Drosophila* nervous system, and are dynamically expressed in unique combinations in larval motoneurons, sensory neurons and muscle tissues[56–60]. This intricate protein interaction network has been shown to provide selective adhesion within the developing and mature *Drosophila* neural network. Notably, in the olfactory system, cell-type specific DIP-Dpr interactions support axonal self-adhesion and the precise sorting of olfactory receptor neurons into distinct territories[60]. In the visual system, interactions between DIP-α, Dpr6 and Dpr10 have been shown to promote cell survival, control layer-specific innervation and modulate synapse density[61]. In addition, DIP-β controls synaptic partner selection by establishing preferences for synapse formation between specific neuron types[62]. More broadly, recent work has shown that gradients of recognition molecules, including DIP-ε and Dpr13, drive a graded pattern of synaptic connectivity in this system[63]. In the neuromuscular system, the interaction between DIP-α and Dpr10 plays a pivotal role in controlling the terminal branching of motoneurons onto larval body wall muscles and adult leg muscles, following specific patterns of innervation at both stages of development[57,59]. The combinatorial expression of DIPs and Dprs in motoneurons and their target muscle further suggests their involvement in regulating neuromuscular connectivity patterns. However, whether DIPs and Dprs function as bona fide synapse organizers in *Drosophila*, and the precise molecular mechanisms underlying their function in synaptic specificity, remain unclear.

There are five IgLON paralogs in mammals: IGLON5, LSAMP (Limbic System Associated Membrane Protein), OPCML (Opioid binding Protein/Cell adhesion Molecule Like), NTM (NeuroTriMin) and NEGR1 (Neuronal Growth Regulator 1). Proteomics studies of the adult mouse brain indicate that IgLONs are expressed in neurons, with some being expressed at high levels, while others are also expressed in glia[64]. In addition, IgLONs exhibit specific, regionally restricted patterns of expression in different brain regions[65]. The early onset of IGLON expression during embryogenesis, which persists into adulthood, suggests diverse functional roles throughout life[66]. Consistent with their distinct expression patterns, knock-out mouse models targeting each of the five IgLON genes exhibit significant but partially overlapping brain dysfunctions[67]. IgLONs are also widely expressed in the human brain and have been associated to a wide spectrum of neurological, neurodevelopmental, neuropsychiatric and neurodegenerative disorders, often involving synaptic dysfunction. These disorders include autoimmune encephalitis, mental retardation, autism spectrum disorders, major depressive disorder, schizophrenia and Alzheimer's disease[67–70]. Paradoxically, the role of IgLONs in mammals is poorly understood.

IgLONs are capable of multiple homomeric and heteromeric interactions with varying affinities between members of the family[30,33–35]. Based on their neuronal expression, ability to engage in trans-cellular interactions, and co-expression in some brain areas, it has been widely suggested that IgLONs could bridge neuronal membranes at synapses and act as synaptic organizers[30]. Accordingly, a few studies support a synaptic role for IgLONs. Notably, proteomic characterization of rodent synaptic clefts has detected all IgLONs at glutamatergic synapses, and IGLON5 at GABAergic synapses as well[71–73]. At the ultrastructural level, LSAMP, OPCML, and NEGR1 have been shown to accumulate at synapses in various developing and adult brain regions[74,75]. In addition, OPCML has been identified at hippocampal synapses and found to promote dendritic stability through ephrin-cofilin signaling and regulation of F-actin dynamics[76]. Finally, a limited number of reports suggest that IgLONs modulate synaptic density in hippocampal neuronal cultures upon overexpression[77,78]. Despite compelling evidence for a role of IgLONs at synapses, this field remains largely unexplored.

Our current data show a pivotal role for the trans-synaptic interaction of *C. elegans* IgLONs in synaptic assembly. In addition, these results demonstrate that IgLONs are able to localize neurotransmitter receptors at the synapse via an unprecedented mechanism involving a direct interaction between an Ig-fold domain of a synaptic adhesion molecule and an ionotropic receptor. This finding is particularly significant given that the Ig superfamily, a large family of single-pass and GPI-anchored cell adhesion molecules, is widely present at synapses and may similarly recruit postsynaptic receptors[79]. Interestingly, a recent study reports the direct binding of two rodent IgLONs (OPCML and NTM) to AMPA receptors. This interaction may occur at synapses and regulate AMPAR mobility. However, the precise binding interface has not yet been identified[80].

The anatomical simplicity of the *C. elegans* nervous system, together with its known inter-individual reproducible connectivity, provides unique opportunities to analyze synapses in vivo at individual resolution. In this study, we have analyzed a subset of synapses formed on the AVA and DB neurons that contain the ACR-16 AChR. AVAs are a pair of "command interneurons" that integrate numerous sensory inputs, connect directly to motoneurons and regulate locomotion[21,81]. These neurons are among the most connected neurons in *C. elegans*, receiving more than 200 inputs, primarily from sensory neurons and other interneurons. However, our results have identified a specific subset of AVA synapses containing RIG-5, ZIG-8 and ACR-16. Overall, our data highlight a strong specificity in the molecular composition of synapses among AVA neurons, raising the question of how RIG-5 and ZIG-8 are trapped at specific synapses.

A first level of control relies on the specific expression of *rig-5* and *zig-8* in defined classes of neurons (Supplementary Fig. 1c). For example, *rig-5* is expressed in the presynaptic AVE neurons but not in the AVB neuron, a cholinergic interneuron that provides 20–30 synaptic inputs but does not use ACR-16 as a receptor at these synapses (Supplementary Fig. 1g). However, cell-specific expression does not explain how RIG-5 and ZIG-8 cluster in specific locations. These molecules are bound to membranes by a lipid anchor and are likely to diffuse freely along the plasma membranes. During development, transient contact between pre- and postsynaptic membranes may be sufficient to engage RIG-5 and ZIG-8 interactions in *trans*, and nucleate synaptic differentiation. Accordingly, no cluster of RIG-5 or ZIG-8 is observed when either molecule is absent (Fig. 2c, d, Supplementary Fig. 4c, d).

Additionally, our data raise the question of how the concentration of RIG-5 and ZIG-8 is achieved. While synaptic scaffolding molecules

have often been shown to multimerize in different stoichiometric complexes[6,8], structural studies of *C. elegans* RIG-5 and ZIG-8, *Drosophila* DIPs and Dprs, and mammalian IGLONs suggest that they solely form dimers through their Ig1 domains, rather than higher-order multimers[30,31,33–35]. Nevertheless, our structural prediction is consistent with each molecule of ZIG-8 interacting in trans with RIG-5, and laterally *in cis* with each subunit of ACR-16, potentially forming a pentameric RIG-5−ZIG-8−ACR-16 assembly (Fig. 5b). This model suggests that ACR-16 could stabilize the whole complex through pentamerization. Accordingly, in the *acr-16* mutant, GFP-RIG-5 and GFP-ZIG-8 display reduced synaptic concentration and a diffuse pattern between synapses, suggesting stabilization defects (Fig. 3d, e).

Although our simple model of protein interactions between IgLONs and ACR-16 is sufficient to explain the formation of postsynaptic AChR clusters, the lack of any transmembrane and intracellular region is difficult to reconcile with coupling to presynaptic differentiation. A recent analysis of the nematode extracellular interactome identified several RIG-5 and ZIG-8 interacting partners, including potential presynaptic proteins[82]. It will be interesting to test these candidates and possibly identify new binding partners of RIG-5 and ZIG-8 to account for the coordinated differentiation of pre- and postsynaptic specializations.

In any case, our genetic strategy in *C. elegans* has uncovered original functions for IgLON molecules and a unique paradigm for AChR clustering. Since all these molecules and protein domains have been conserved during evolution, it will be interesting to test to what extent these mechanisms are utilized in the mammalian brain.

## Methods

### Strains
All experiments were performed at 20 °C. Strains were grown on nematode growth medium (NGM) agar plates seeded with *Escherichia coli* OP50[83]. The wild-type reference strain was *C. elegans* N2 Bristol. All strains and alleles used in this study are described in Supplementary Table S1 and S2, respectively.

### Genetic screen
**Mutagenesis.** Mutagenesis was performed using the EN7643 strain (*krSi81[Pmyo-3::TIR1::bfp]; kr463[acr-16::aid::scarlet]*). Animals were exposed to 47 mM ethyl methanesulfonate (EMS; Sigma) for 4 h at 20 °C. Following treatment, populations were maintained at 15 °C, 20 °C, or 23 °C to promote staggered hatching. A total of five independent mutagenesis experiments were conducted, and 12,912 *C. elegans* haploid genomes were screened.

**Isolation of mutants.** Clonal F2 progeny of mutagenized EN7643 animals, grown on auxin-supplemented NGM plates, were examined using a Nikon AZ100 Multizoom microscope. Animals exhibiting abnormalities in the ACR-16::AID::Scarlet expression pattern were isolated and categorized into four phenotypic classes: absence of signal, weak puncta, diffuse signal, and other distribution defects. Mutants exhibiting diffuse signal along neurites were prioritized, as this phenotype was consistent with impaired ACR-16 clustering. Animals with homozygous mutations were isolated in the F3 progeny.

**Mapping, whole genome sequencing and validation of mutations.** Homozygous mutant strains were backcrossed twice to the parental EN7643 strain. After the second backcross, five independent F2 recombinant mutants were pooled for genomic DNA extraction and whole-genome sequencing (Eurofins and Novogene). A local bioinformatics pipeline derived from the CloudMap pipeline was used to identify candidate causative mutations and regions of genetic linkage, indicated by clusters of EMS-induced variants across large chromosomal segments[84,85]. Candidate mutations were validated by recreating the mutation in the parental EN7643 background and/or by reverting

the mutation to the wild-type sequence in the mutant strain. Of the 53 mutant strains isolated, 12 exhibited a diffuse ACR-16::Scarlet distribution phenotype. These included five independent alleles of *rig-5* and *zig-8: rig-5(kr573[Q16*]), rig-5(kr849[P191S]), zig-8(kr850[Q92*]), zig-8(kr851[A68T])* and *zig-8(kr852[I239F])* (Supplementary Fig. 2).

### Allele generation by CRISPR/Cas9 genome engineering
All knock-in alleles were generated according to a standard protocol[86]. crRNA was designed using Benchling software and synthesized by Integrated DNA Technologies (IDT). To form a triplex, a mix of 2.8 µL crRNA (34 µM), 5 µL tracrRNA (18 µM) and 0.5 µL Cas9 nuclease (10 µL/µg) (IDT) was incubated for 15 min at 37 °C. For deletions, two crRNA were used at both ends of the target sequence, with a triplex made of 1.4 µL of each crRNA. A mix containing 2.2 µL single-strand repair template (1 µg/µl) or 500 ng double-strand repair template, with 800 ng of pRF4 plasmid, and molecular biology grade water up to 20 µL was added to the triplex. This mix was injected into adult *C. elegans*. Worms with a roller phenotype in the next generation were isolated and tested by PCR. All gene edits were confirmed by sequencing. Finally, the edited strains were outcrossed once to remove nonspecific background mutations. A list of crRNA is provided in Supplementary Table S3.

### Plasmids
The plasmids constructed for this study are described in Supplementary Table S4. All constructs were verified by Sanger sequencing from the GATC company. Plasmid sequences are available upon request.

### Tissue-specific expression
For tissue-specific expression, the promoters used were as follows:
- *Pmyo-3*: body wall muscle cells
- *Punc-17*: cholinergic motoneurons
- *Punc-129*: D-type motoneurons (DA and DB type)
- *Ppept-3*: AVE neurons[87]

For specific expression in AVA and AVB neurons, the Cre-loxP recombination strategy was used, with promoter combination as follows: *Pflp-18* and *Pgpa-14* (AVA neurons), *Ptkw-40* and *Plgc-55* (AVB neurons)[25,88,89]. Promoter boundaries are described in Supplementary Table S5.

The protein sequences of *rig-5* and *zig-8* are provided in Supplementary Table S7. The cDNAs were cloned by RT-PCR from mixed-stage *C. elegans* RNAs.

### Generation of single-copy insertion alleles
Single-copy insertion alleles were generated by the miniMos method[90]. Briefly, adult *C. elegans* were injected with a mix containing 15 ng/µL of the plasmid of interest (containing promoters and open reading frames fused to fluorescent proteins), 50 ng/µL pCFJ601 (Mos1 transposase), 10 ng/µL pMA122 (negative selection marker *Phsp16.2::peel-1*), and 2.5 ng/µL pCFJ90 (*Pmyo-2::mCherry*). Neomycin (G418) was added to plates 24 h after injection at 1.5 µg/µL final concentration. Candidate plates were heat-shocked for 2 h at 34 °C. Worms with an insertion were isolated and homozygosed.

### Multicopy and extrachromosomal array lines
Extrachromosomal array alleles were generated by injecting a mix of plasmids of interest as depicted in the Supplementary Table S2.

For *acr-16* transcriptional reporter (*krIs83[Pacr-16::aid::mNG]*, Supplementary Fig. 1b), a strain bearing a *Pacr-16::aid::mNG* extrachromosomal array was randomly integrated by X-ray irradiation (40 Gy).

### Auxin-induced cell specific degradation
For all images except in Fig. 4e, f, Fig. 6, Supplementary Fig. 1b, Supplementary Fig. 3a, Supplementary Fig. 4 and Supplementary Fig. 5,

worms were grown on auxin plates to degrade ACR-16-AID-Scarlet at NMJs. To induce protein degradation, adult *C. elegans* were transferred to auxin plates, and their progeny was analyzed. Auxin plates were prepared by adding auxin (indole-3-acetic acid, Sigma-Aldrich) from a 400 mM stock solution in ethanol to NGM, at the final concentration of 1 mM[22].

## Neuron-type Specific Illumination (NeuroSIL)

ACR-16 was tagged with AID and Scarlet sequences, along with three copies of spGFP11 (ACR-16-AID-Scarlet-spGFP11). The spGFP1-10 moiety was expressed in the cytoplasm of AVA or DB neurons using selected combinations of promoters, as described above. All ACR-16 clusters displayed Scarlet labelling but only ACR-16 clusters formed in the AVAs or DBs, exhibited GFP fluorescence upon reconstruction. Two to four transgenic lines were observed, and/or used for quantification: Fig. 1e (2 lines), Fig. 1h (4 lines) and Supplementary Fig. 1f (2 lines).

## Microscopy imaging and quantification

For spinning disk imaging, young adult hermaphrodites were used, except in Supplementary Fig. 4, in which L1 larvae were imaged 1-2 h after hatching. Live worms were mounted on 2% agarose dry pads with 2% (for young adults) or 5% (for L1 larvae) polystyrene beads (Polybeads, Polysciences) in M9 buffer. Worms were observed using an Andor spinning disk system (Oxford Instruments) installed on a Nikon-IX86 microscope (Olympus) equipped with a 60×/NA 1.42 oil-immersion objective and an Evolve electron-multiplying charge-coupled device camera. Each animal was imaged with IQ software (APIS Informationstechnologien) or Metamorph software (version 7.10.5.476) as a stack of optical sections (0.2 μm apart) across the whole thickness of the ventral or dorsal nerve cords. All images were processed using Fiji software (v2.0) and correspond to the sums of all slices with signal along the *Z* axis[91,92].

Acquisition settings were the same across genotypes for quantitative analysis. Image quantification was performed using a macro in Fiji. For total fluorescence intensity measurements, a 50 μm (wide) × 3 μm (high) rectangle containing solely the nerve cord (first quarter) was cropped and projected as a sum in Z. The mean intensity was then projected in the x direction (orthogonal to the cord), and the intensity was measured as the area under the curve, after background exclusion[93]. Data are presented as a percentage of the average fluorescence relative to that of the wild type.

For puncta density and intensity quantification, a CORSEN analysis was used in Fig. 3d, e[94]. Due to the diffuse localization pattern observed in Fig. 2c–e, Fig. 3a–c, Fig. 5a, e, puncta density was quantified as follows: the cropped image (50 μm wide × 3 μm high) was converted to a one-dimensional axis along the y axis in Fiji (ImageJ). Plot profiles were generated for each image and saved as csv files. To determine puncta density, the csv files were computed by an R script to apply a threshold, define the background signal and count the number of peaks above this threshold[95]. The threshold was the same across genotypes for each quantitative analysis.

In Figs. 1c and 4a-d, the total number of puncta along the ventral nerve cord was quantified manually. For rescue experiments, one to three lines transgenic lines were observed, and/or used for quantification: Fig. 4a (1 line), Fig. 4b (1 line), Fig. 4c (2 lines), Fig. 4e (3 lines) and Fig. 4f (2 lines). Quantifications were performed on a single day to ensure consistency, whereas qualitative assessments of the rescues were repeated on three separate days to confirm reproducibility. The quantification of BFP-tagged proteins could not be assessed automatically at neuron-neuron synapses due to a low signal-to-noise ratio. For microscopy images show in Fig. 1b, h; 2a, b and Supplementary Fig. 1b, g-h; 3b; 4a-d, the observations were repeated on three different days.

For colocalization analysis, images were captured along 50 μm of the dorsal nerve cord anterior to the vulva. Fluorescence intensity along the cord was evaluated with the Plot Profile Fiji plugin. For each channel, values along the x axis were normalized to the maximal intensity value. Data are presented as minimum to maximum values for animals of each genotype. Colocalization between two channels was analyzed using Pearson's correlation coefficient as previously described[18]. For quantification, data are presented as box plots showing lower and upper quartiles (box), mean (square), median (center line) and standard deviation (whiskers). Statistical tests were performed using RStudio version 2022.12.0 (R version 4.0.5).

Image quantification for each experiment was performed on the same day using standardized acquisition settings, thereby ensuring accurate and comparable measurements across genotypes. Consistency of phenotypes among genotypes was verified across at least three independent days of observation, confirming that the observed results were reproducible. All attempts at replication were successful. Blinding was not performed because data analysis was automated through macros applied to large datasets, minimizing the potential for user bias.

The NeuroPAL (Neuronal Polychromatic Atlas of Landmarks) strain was used to identify *acr-16*-expressing neurons based on a *Pacr-16::aid::mNG* transcriptional reporter in head, body and tail regions, by comparison to the reference manual[23].

## Structure prediction and analysis

Protein complex structure predictions were performed with Alpha-Fold-Multimer, using the ColabFold implementation version 1.5.2 running Alphafold release 2.3.1[96–98]. For the prediction, we used the default settings, including the use of template information, relaxation of the predicted structures using amber force fields, and use of both paired and unpaired multiple sequence alignments. We ran alphafold predictions of the ACR-16 pentamer, and various ectodomain complexes of ACR-16 and ZIG-8, all returning models that were in close agreement. Alphafold reported high confidence values for the interfaces: the iPTM value for the ACR-16 pentamer, the 1:1 complex of ACR-16 and ZIG-8 and the ACR-16 pentamer with a ZIG-8 molecule were 0.849, 0.865 and 0.845. An example of the sequence coverage, pLDDT, and predicted alignment error (PAE) plots are shown in Supplementary Fig. 7b. Analysis of the structures and creation of composite models (including the ZIG-8-RIG-5 crystal structure) were performed using the molecular graphics program PyMOL (v. 3.0.4)[31,99].

## Locomotion test

Worm locomotion was assessed on plates by gently tapping the head and tail of each animal using a platinum wire mounted on a pick, as well as following mechanical tapping of the Petri dish. The assessments were conducted by three experimenters who were blinded to the genotype.

## Reporting summary

Further information on research design is available in the Nature Portfolio Reporting Summary linked to this article.

# Data availability

Source data are provided with this paper. The plasmids and *C. elegans* strains generated in this study are available from the corresponding authors upon request. Source data are provided with this paper.

# Code availability

The code used to perform analyses of synapse density is deposited on Zenodo, and is accessible via (https://doi.org/10.5281/zenodo.17571550)[95].

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

## Acknowledgements

We thank members of the Bessereau lab for providing feedback, Mélissa Cizeron for advice on image analysis, Driss Laabid and Lina Boumasmoud for technical assistance, and Océane Romatif, Delphine Le Guern and Camille Vachon for strains. We are grateful to Elise Forgues, Cecile Chatras, Mathieu Ben Abu and Manon Courtieux for their help during their respective internships. We thank Pierre-Jean Corringer for critical reading of the manuscript, Mei Zhen and Jun Meng for providing information on neuronal promoters, Alexander Gottschalk for sharing plasmids and Eviatar Yemini for advice on NeuroPAL. We thank the Caenorhabditis Genetics Center (CGC), funded by NIH Office of Research Infrastructure Programs (P40 OD010440), for providing strains. We thank the SFR Biosciences (University Lyon 1 CNRS UAR 3444 INSERM US8, ENS de Lyon), Matthieu Caron and Francesca Palladino for access to equipment. We thank Le Centre d'Imagerie Quantitative Lyon-Est (LyMIC-CIQLE, Lyon, France) imaging facility for support and access to equipment, and Camilla Luccardini for technical assistance. Some strains were generated by SEGiCel (SFR Santé Lyon Est CNRS UAR 3453, Lyon, France) with the support of CNRS and IBiSA. Some graphical elements in Figs. 1d and 6b were provided by Servier Medical Art (https://smart.servier.com), licensed under CC BY 4.0 (https://creativecommons.org/licenses/by/4.0/). This work was supported as follows: M.M.: fellowship from the French Ministry of Research; BPL: LABEX Cortex (ANR-11-LABX-0042) of University Lyon 1, within the program "Investissements d'Avenir" (ANR-11-IDEX-0007) and Fondation pour la Recherche Médicale grant (FRM-MND-202411019867); EÖ: National Institutes of Health, National Institute of Neurological Disorders and Stroke grant R01 NS139060 grant; JLB: Équipe FRM 2023 EQU202303016267, ANR Synapunct ANR-22CE16-0024-01, ERC_Adg C.NAPSE #695295, ANR-11-LABX-0042/ANR-11-IDEX-0007.

## Author contributions

Conceptualization: M.M., J.-L.B. and B.P.-L.; Methodology: M.M., A.W., E.Ö., J.-L.B. and B.P.-L.; Investigation: M.M., L.P., L. G. and B.P.-L.; Formal analysis: M.M., L.P., E.Ö. and B.P.-L.; Visualization: M.M., E.Ö., J.-L.B. and B.P.-L.; Writing – Original Draft: M.M., J.-L.B. and B.P.-L.; Writing – Review & Editing: M.M., A.W., E.Ö., J.-L.B. and B.P.-L.; Funding Acquisition: E.Ö., J.-L.B. and B.P.-L.; Supervision: J.-L.B. and B.P.-L.

## Competing interests

The authors declare no competing interests.
