## [Transparent Peer Review file · Nature Communications]

A trans-synaptic IgLON adhesion molecular complex directly contacts and clusters a nicotinic receptor

Corresponding Author: Dr Berangere Pinan-Lucarre

Version 0:

Reviewer comments:

Reviewer #1

(Remarks to the Author)

In this study, Mialon et al. identify a novel mechanism for the clustering of acetyl choline receptors (AChRs) at neuronal synapses. Using a series of sophisticated manipulations in *C. elegans*, they convincingly show that extracellular interactions mediated by adhesion molecules of the IgLON family localize the ACR-16 AChR at the synapse. They find that RIG-5, a presynaptic IgLON, and ZIG-8, a postsynaptic IgLON, interact via their N-terminal Ig1 domains to form a trans-synaptic complex. In addition, ZIG-8 interacts in cis with ACR-16 via its Ig2 domain. Using AlphaFold multimer modelling, the authors identify a single amino acid substitution in ACR-16 that is sufficient to perturb its interaction with the ZIG-8 Ig2 domain, resulting in a loss of ACR-16 clustering at synapses. Finally, the authors show that the interaction with the ZIG-8 Ig2 domain is sufficient to mislocalize ACR-16 to GABAergic postsynaptic compartments. Taken together, this study identifies a novel mechanism for localizing AChRs at synapses, which may have implications for mammalian nervous systems. The rationale for the experiments is clearly explained, results are clearly described, elegant assays such as extra- and intracellular split-GFP complementation are used, and the experiments are performed to a high standard. I have only one major comment and several minor ones for improvement of the manuscript.

Major comment

1. The authors test the interaction between ACR-16 or ACR-16(T201R) and ZIG-8 in a co-immunoprecipitation assay in HEK293T cells (Fig. 5B). The blot shown is not entirely convincing and appears to have some smudges on the left side of the RFP blot. The IP of ACR-16 in the condition where it is coexpressed with ZIG-8 (right lane) is weak, resulting in very low levels of coimmunoprecipitated ZIG-8. This blot should be repeated to convincingly show the interaction between ACR-16 and ZIG-8, and absence of the interaction between ACR-16(T201R) and ZIG-8. In addition to showing the pulldown lanes, the authors should also include the input lanes, to show presence of ACR-16 and ZIG-8 across the various conditions.

Minor comments

1. Mutations in *rig-5* and *zig-8* were identified in a genetic screen. From supplementary table S2 it was not clear to me whether these were the only candidates, or top candidates, or two of multiple candidates, to affect ACR-16 localization. A bit more background on this screen and the identification of *rig-5* and *zig-8* would be helpful.
2. Fig. 4D (cell type-specific rescues) shows that expression of ZIG-8 in the AVA neurons leads to a strong increase in apparent synaptic puncta density. The reason for this is not entirely clear. From the image in Fig. 4B, there actually appears to be a decrease in ZIG-8 puncta density and an increase in diffuse ZIG-8 signal. Is the increase in puncta density due to the more diffuse distribution of ZIG-8? Are these additional clusters synaptic? Or could this indicate that ZIG-8 can induce additional ACR-16 clusters, partly independent of presynaptic RIG-5? Or could the apparent increase in ACR-16 clusters be due to the presence of RIG-5 in other presynaptic neurons besides the AVE neurons? Please discuss.
3. Related to the comment above: the number of ACR-16 puncta in AVE-to-AVA synapses is ± 16 in Fig. 1C and ± 29 in Fig. 4D (rescue with presynaptic RIG-5 in AVE and postsynaptic ZIG-8 in AVA neurons). Please comment on these differences.
4. There are many AChRs in *C. elegans*, but only 2 IgLON orthologs. Do the authors think the interaction with IgLONs might also occur with other AChRs or is this specific for ACR-16? Is the interacting region in ACR-16 (T201) conserved in other AChRs?
5. Any insight on whether RIG-5 and ZIG-8 affect AChR function would further strengthen the manuscript. I understand electrophysiology is not trivial, but possibly other approaches exist? E.g. does AlphaFold multimer modelling show changes in AChR structure in the presence of ZIG-8?

Reviewer #2

(Remarks to the Author)

Precise subcellular localization and clustering of NTRs at post-synaptic sites is crucial for proper neuronal function, but how this is achieved, and specifically the roles that extracellular proteins play in this process, is poorly understood. In *C.elegans*, Ach is the most widely used NT. ACR-16 (the $\alpha 7$ -like AChR subunit), was previously shown to cluster in the NMJ by intracellular protein scaffolds, but how it clusters in neuron-neuron synapses remains unknown.

In this study, Mialon and colleagues focus on specific neurons in the worm nerve cord, and find that the IgLON homologs RIG-5 and ZIG-8 localize to pre- and postsynapses, respectively, and trans-neuronally interact (via Ig1) to control the localization of one another. Moreover, ZIG-8 directly binds and recruits ACR-16 via its Ig2 domain. Thus, they uncover a novel mechanism controlling the synaptic localization of an AChR via extracellular interactions with IgSF synaptic molecules.

This is a very elegant study, well-constructed and addressing the majority of the critical aspects required for the authors to establish their hypothesis. It is carefully designed and highly convincing, and its potential implications on other neuronal systems and higher organisms – given the conserved nature of all molecules involved – is of a broad interest to the molecular neuroscience community. I actually think that if the authors would have written the paper in a bit more broader manner, and perhaps focusing on synaptic targeting rather than NTR localization, this study could be published in the top journals. That said - in its current state and with minor textual changes and improvements of figure presentation - this is a top paper for Nature Comm.

- 1) Annotation of the figures in many places is lacking and they are sometimes confusing or difficult to interpret (in some cases even after reading the entire legend). The system is not straightforward and involves many types of neurons that is difficult for the non-worm expert to follow. There are some schemes throughout which try to clarify, but they are not clear enough (e.g., 1D doesn't show AVE, 1F is tiny and hard to understand - zoom in would help). Better annotation of the images themselves would also be helpful (e.g. S Fig. 5, and other places - two types of arrows indicate two different neurons - but the legend doesn't say which type points to what).
- 2) The intro is, in my opinion, a bit tedious. Too many facts and gene names. It's a stylish preference - so completely up to the authors - but one could make the paper lighter and more interesting to the general reader.
- 3) Tissue specific manipulation - again worm specific, but I could not find specifics on how the authors control the tissue-specificity of the AID experiments (Fig. 1B) – how is the auxin administered/introduced to the animal?
- 4) The screen - no hard data or information is provided about the genetic screen in which zig8 and rig5 were identified.
- 5) "dimmer" puncta in 3E - not evident in the fluorescent images.
- 6) "Taken together, our data strongly suggest that the main function of RIG-5 and ZIG-8 is to cluster the nicotinic receptor ACR-16 at specific cholinergic synapses." - I find this a bit too strong of a statement.
- 7) In figure 4 - what does the change in the number of puncta mean? this is not really evident in the fluorescent images BTW. Also - how does the number of puncta relate to the number of synapses (in figure 1) - this is unclear. ALSO - if Zig8 and Rig5 interact independently of Acr16 then why does it matter which of the proteins is in the pre- or post- synapse? This relates to 4E/F
- 8) coIP does not equal with physical interaction but rather presence in the same complex.
- 9) are these worms "normal"?

I want to reiterate - this is an excellent and very elegant paper. One of its main discoveries is the way by which igSF proteins 'match' pre and post synaptic partners - which is by clustering neurotransmitter receptors. Super cool, super relevant - definitely accept.

Version 1:

Reviewer comments:

Reviewer #1

(Remarks to the Author)

I appreciate the extensive effort the authors have put in to further demonstrate the interaction between ZIG-8 and ACR16 and agree with their decision to remove the western blot data from the manuscript - the interaction is convincingly supported by other lines of evidence. The added clarifications in response to the reviewers' comments have further improved an already excellent and exciting study - this manuscript is ready for publication.

Reviewer #2

(Remarks to the Author)

In this revised version of the manuscript the authors addressed most of my comments of what I already thought was a very good manuscript. I do have only one suggestion - the title of the last section of the results "ZIG-8 Ig2 domain binds the ACR-16 extracellular domain" is not, in my opinion, totally supported by the data. Is it the most plausible? likely yes. But I don't think - especially now without the coIP (and even with it) that the authors can truly make this claim. If you want to maintain this claim then perform SPR experiments on isolated proteins. This is not necessary in my opinion but then tweak down this

statement.

Response to reviewers

A trans-synaptic IgLON adhesion molecular complex directly contacts and clusters a nicotinic receptor

REVIEWER COMMENTS

Reviewer #1 (Remarks to the Author):

In this study, Mialon et al. identify a novel mechanism for the clustering of acetyl choline receptors (AChRs) at neuronal synapses. Using a series of sophisticated manipulations in *C. elegans*, they convincingly show that extracellular interactions mediated by adhesion molecules of the IgLON family localize the ACR-16 AChR at the synapse. They find that RIG-5, a presynaptic IgLON, and ZIG-8, a postsynaptic IgLON, interact via their N-terminal Ig1 domains to form a trans-synaptic complex. In addition, ZIG-8 interacts in cis with ACR-16 via its Ig2 domain. Using AlphaFold multimer modelling, the authors identify a single amino acid substitution in ACR-16 that is sufficient to perturb its interaction with the ZIG-8 Ig2 domain, resulting in a loss of ACR-16 clustering at synapses. Finally, the authors show that the interaction with the ZIG-8 Ig2 domain is sufficient to mislocalize ACR-16 to GABAergic postsynaptic compartments. Taken together, this study identifies a novel mechanism for localizing AChRs at synapses, which may have implications for mammalian nervous systems. The rationale for the experiments is clearly explained, results are clearly described, elegant assays such as extra- and intracellular split-GFP complementation are used, and the experiments are performed to a high standard. I have only one major comment and several minor ones for improvement of the manuscript.

Major comment

1. The authors test the interaction between ACR-16 or ACR-16(T201R) and ZIG-8 in a co-immunoprecipitation assay in HEK293T cells (Fig. 5B). The blot shown is not entirely convincing and appears to have some smudges on the left side of the RFP blot. The IP of ACR-16 in the condition where it is coexpressed with ZIG-8 (right lane) is weak, resulting in very low levels of coimmunoprecipitated ZIG-8. This blot should be repeated to convincingly show the interaction between ACR-16 and ZIG-8, and absence of the interaction between ACR-16(T201R) and ZIG-8. In addition to showing the pulldown lanes, the authors should also include the input lanes, to show presence of ACR-16 and ZIG-8 across the various conditions.

We agree with our reviewer that the technical quality of the co-IP is not optimal. This is largely due to the intrinsic difficulty of expressing AChRs in heterologous systems, and in particular ACR-16, which requires the co-expression of the RIC-3 chaperone. Despite numerous attempts to optimize transfection conditions, expression times, and plasmid combinations, we were unable to obtain convincing co-immunoprecipitation results due to persistent expression issues. We therefore turned to an alternative strategy and invested considerable effort in developing a flow cytometry-based assay. We attempted to produce ACR-16 in insect cells and generated recombinant ZIG-8 fused to streptavidin, which was further detected by fluorescent biotin. Unfortunately, once again, ACR-16 expression was too low to yield a signal above background.

Since we believe that the ZIG-8—ACR-16 interaction is strongly supported by *in silico* evidence (AlphaFold-Multimer iPTM score >0.8), point mutagenesis (disruption of ACR-16 (T201R) clustering at neuronal synapses (Fig. 5e) but not at NMJs (Sup Fig. 8)), and the sufficiency of the ZIG-8 Ig2 domain for ACR-16 clustering when ectopically expressed in muscle (Figure 6c-e), we have decided to remove the co-IP data from the revised manuscript. We have therefore removed panel B from Figure 5 and renamed the remaining panels. We have removed the following text from the main text:

“Our results were compatible with a direct interaction between ZIG-8 and ACR-16. To test this hypothesis, we performed a co-immunoprecipitation experiment in HEK cells expressing both proteins. To prevent ZIG-8 from being released into the culture medium via cleavage of the GPI anchor, we replaced the C-terminal sequence of ZIG-8 containing the GPI anchor site with the transmembrane domain of human CD4 (GFP-ZIG-8-CD4(TM)). In addition, we coexpressed the RIC-3 chaperone to ensure proper intracellular trafficking of ACR-16-Scarlet³⁸. The pull down of ACR-16-Scarlet efficiently co-immunoprecipitated GFP-ZIG-8-CD4(TM), hence supporting the existence of a physical interaction between ZIG-8 and ACR-16 (Figure 5B).”

“Consistently, the ACR-16(T201R)-AID-Scarlet receptor failed to co-immunoprecipitate GFP-ZIG-8-CD4(TM) in the HEK cell assay (Figure 5B).”

Minor comments

1. Mutations in *rig-5* and *zig-8* were identified in a genetic screen. From supplementary table S2 it was not clear to me whether these were the only candidates, or top candidates, or two of multiple candidates, to affect ACR-16 localization. A bit more background on this screen and the identification of *rig-5* and *zig-8* would be helpful.

We thank the reviewer for the opportunity to elaborate on our genetic screen. In response, we have added a dedicated paragraph in the Methods section (line 816). Major changes are highlighted in yellow in the article file.

“Genetic screen

Mutagenesis. Mutagenesis was performed using the EN7643 strain (*krSi81[Pmyo-3::TIR1::bfp]; kr463[acr-16::aid::scarlet]*). Animals were exposed to 47 mM ethyl methanesulfonate (EMS; Sigma) for 4 hours at 20 °C. Following treatment, populations were maintained at 15 °C, 20 °C, or 23 °C to promote staggered hatching. A total of five independent mutagenesis experiments were conducted, and 12,912 *C. elegans* haploid genomes were screened.

Isolation of mutants. Clonal F2 progeny of mutagenized EN7643 animals, grown on auxin-supplemented NGM plates, were examined using a Nikon AZ100 Multizoom microscope. Animals exhibiting abnormalities in the ACR-16::AID::Scarlet expression pattern were isolated and categorized into four phenotypic classes: absence of signal, weak puncta, diffuse signal, and other distribution defects. Mutants exhibiting diffuse signal along neurites were prioritized, as this phenotype was consistent with impaired ACR-16 clustering. Animals with homozygous mutations were isolated in the F3 progeny.

Mapping, whole genome sequencing and validation of mutations. Homozygous mutant strains were backcrossed twice to the parental EN7643 strain. After the second backcross, five independent F2 recombinant mutants were pooled for genomic DNA extraction and whole-genome sequencing (Eurofins and Novogene). A local bioinformatics pipeline derived from the CloudMap pipeline was used to identify candidate causative mutations and regions of genetic linkage, indicated by clusters of EMS-induced variants across large chromosomal segments⁸³⁻⁸⁴. Candidate mutations were validated by recreating the mutation in the parental EN7643 background and/or by reverting the mutation to the wild-type sequence in the mutant strain. Of the 53 mutant strains isolated, 12 exhibited a diffuse ACR-16::Scarlet distribution phenotype. These included five independent alleles of *rig-5* and *zig-8*: *rig-5(kr573[Q16*])*, *rig-5(kr849[P191S])*, *zig-8(kr850[Q92*])*, *zig-8(kr851[A68T])* and *zig-8(kr852[I239F])* (Supplementary Figure 2).”

2. Fig. 4D (cell type-specific rescues) shows that expression of ZIG-8 in the AVA neurons leads to a strong increase in apparent synaptic puncta density. The reason for this is not entirely clear. From the image in Fig. 4B, there actually appears to be a decrease in ZIG-8 puncta density and an increase in diffuse ZIG-8 signal. Is the increase in puncta density due to the more diffuse distribution of ZIG-8? Are these additional clusters synaptic? Or could this indicate that ZIG-8 can induce additional ACR-16 clusters, partly independent of

presynaptic RIG-5? Or could the apparent increase in ACR-16 clusters be due to the presence of RIG-5 in other presynaptic neurons besides the AVE neurons? Please discuss.

We thank the reviewer for this insightful comment. In these rescue experiments, we likely observe distinct classes of synapses depending on the genetic background: AVE > AVA connections in **Figure 4a, c and e**; and synapses onto AVA from various presynaptic neurons—including but not limited to AVE—as shown in **Figure 4b**. Although the number of puncta observed in the rescue conditions (**Figure 4**) is broadly similar to those quantified under physiological conditions, they are not identical (**Figure 1c**). For instance, in **Figure 4b**, where BFP-ZIG-8 is expressed in AVA neurons of a *zig-8* mutant, we counted 80.7 ± 11.7 puncta, whereas one might expect approximately 108 ± 10 synapses based on the endogenous context.

We believe this discrepancy primarily arises from differences in the mode of expression of adhesion molecules. In contrast to the fluorescent knock-in reporters used throughout the study, the rescue constructs rely on transgenic overexpression driven by cell-specific promoters (*Pept-3* for AVE neurons or a combination of *Pgpa-14* and *Pflp-18* for AVA neurons). Furthermore, in **Figure 4b**, ZIG-8 is tagged with a blue fluorescent protein (tagBFP), which is inherently dimmer and more difficult to detect than commonly used green or red fluorescent proteins. This may contribute to the more diffuse appearance of the BFP-ZIG-8 signal.

In principle, a more physiological approach to dissect cell-specific requirements could involve auxin-mediated degradation of endogenous proteins. However, this strategy depends on cytosolic degron tagging, which is not feasible for GPI-anchored proteins such as ZIG-8 and RIG-5. Consequently, cell-specific rescue using transgenic expression was the most viable option for these experiments.

We have improved **Figure 4**'s display and added two sentences to clarify the experiment (lines 283 and 291):

“In these rescue experiments, we observed distinct classes of synapses depending on the genetic background.”

“We observed an average of 80 puncta containing both BFP-ZIG-8 and ACR-16-AID-Scarlet along the ventral nerve cord. They likely correspond to synapses where ACR-16 clusters in AVA neurons, from various presynaptic neurons, including but not limited to AVE neurons (**Figure 4b, d, Figure 1c, f**).”

Moreover, we have added a discussion of this point in the revised manuscript (line 311) :

“Notably, the number of puncta observed in rescue experiments involving transgenic overexpression is comparable, though not identical, to those quantified using endogenous knock-in reporters. For instance, in the *rig-5; zig-8* double mutant background, expression of BFP-RIG-5 in AVE and Scarlet-ZIG-8 in AVA neurons resulted in 29.9 ± 6.9 AVE-to-AVA puncta (**Figure 4c, d**), whereas under physiological conditions, only 16 ± 3 AVE-to-AVA synapses were observed (**Figure 1c, h**). Such increase is likely caused by transgenic overexpression, suggesting that the precise temporal and quantitative regulation of RIG-5 and ZIG-8 is essential for establishing the correct number of synapses.”

3. Related to the comment above: the number of ACR-16 puncta in AVE-to-AVA synapses is ± 16 in Fig. 1C and ± 29 in Fig. 4D (rescue with presynaptic RIG-5 in AVE and postsynaptic ZIG-8 in AVA neurons). Please comment on these differences.

Please see our previous answer.

4. There are many AChRs in *C. elegans*, but only 2 IgLON orthologs. Do the authors think the interaction with IgLONs might also occur with other AChRs or is this specific for ACR-16? Is the interacting region in ACR-16 (T201) conserved in other AChRs?

The conservation of a ZIG-8-AChR interaction is a very interesting question that we have sought to address. To this end, we performed a Clustal alignment of a ACR-16 region spanning from the β -sheet 7 to loop C with its closest paralogs (ACR-15, -7 and -10)^{PMID: 34388373}. Residues at the predicted interface with ZIG-8 are highlighted in yellow, and the T201 residue is marked in red. This alignment shows poor conservation of the interacting residues, suggesting that the predicted interaction between ZIG-8 and ACR-16 is specific.

```

ACR-16  DEQKCFKFGSWTYDGYKLDLQPA-----TGGFDISEYISNGEWALPLTTVERNEKIFYDCCPEPY 60
ACR-15  DEQVCYFKFGSWTYTRDKIQLE-----KGDFDFSEFIPNGEWIIIDYRTNITVKQYECCPEQY 58
ACR-7   DEQICFMKFGSWTYHGFALDLRLDVV-KGQEPSADLSTYITNGEWHLLSAPARREEKFYKCCPEPY 65
ACR-10  DDQVCYLKFGSWTYHGLALDLSIIAEEDDSELSIDLSTYTPSGEWHLTKAPAVKDVKYNSCCPEPY 66
*: *  *: :*****      ::*          . *: * : .*** : .      * .** * *

```

The interaction between ACR-16 and ZIG-8 was robustly predicted using AlphaFold-Multimer. In principle, a similar strategy could be used to assess potential interactions between ZIG-8 and other AChRs. However, such analyses are complicated by the existence of approximately 30 AChR subunits, which can assemble into heteromeric complexes of up to five distinct subunits, with interaction interfaces possibly spanning multiple subunits.

Even if ZIG-8 does not bind other *C. elegans* AChR through the same interface, this raises the broader issue of whether interactions involving IgLONs and ionotropic receptors may be relevant across species. Interestingly, during the course of our revision, a preprint from the Haganir lab reported binding of two rodent IgLONs to AMPA receptors and suggested a role in modulating receptor mobility. However, the study did not identify the binding interface. We have added a brief discussion on this report (line 503).

“Interestingly, a recent preprint reports the direct binding of two rodent IgLONs (OPCML and NTM) to AMPA receptors. This interaction may occur at synapses and regulate AMPAR mobility. However, the precise binding interface has not yet been identified⁷⁹.”

5. Any insight on whether RIG-5 and ZIG-8 affect AChR function would further strengthen the manuscript. I understand electrophysiology is not trivial, but possibly other approaches exist? E.g. does AlphaFold multimer modelling show changes in AChR structure in the presence of ZIG-8?

Whether ZIG-8 modulates the function of ACR-16 is an intriguing question. We agree with the reviewer that electrophysiological analysis is technically challenging. Moreover, whereas AlphaFold-Multimer can reveal plausible binding modes and suggest how such interactions might alter receptor conformation, it does not directly predict functional outcomes, such as changes in ion conductance, ligand affinity, gating, or desensitization kinetics.

To explore this further, we used AlphaFold-Multimer v3 to model both an ACR-16 homopentamer (magenta) and a complex between the ACR-16 homopentamer and the Ig2 domain of ZIG-8 (shades of green). The two predicted structures were aligned using ChimeraX. The resulting comparison suggests that ZIG-8 binding does not induce substantial conformational changes in the extracellular domain of ACR-16 and the ion pore, nor does it obstruct the agonist-binding pocket located at the top of the receptor.

Given the potential functional relevance of this interaction, we will maintain this brief discussion in our manuscript (line 432):

“Due to their distinct binding specificity to AChRs and unique expression patterns in the mammalian brain, immune system, and other tissues, Lynx prototoxins have been proposed to modulate the function of specific AChR subtypes in a regionally specific manner. However, they have not been shown to control AChR localization. In *C. elegans*, it might be interesting in the future to test whether ZIG-8 binding modulates the function of ACR-16 in a similar manner, although ZIG-8 does not interfere with the ACR-16 agonist-binding site based on *in silico* prediction.”

Reviewer #2 (Remarks to the Author):

Precise subcellular localization and clustering of NTRs at post-synaptic sites is crucial for proper neuronal function, but how this is achieved, and specifically the roles that extracellular proteins play in this process, is poorly understood. In *C. elegans*, Ach is the most widely used NT. ACR-16 (the $\alpha 7$ -like AChR subunit), was previously shown to cluster in the NMJ by intracellular protein scaffolds, but how it clusters in neuron-neuron synapses remains unknown.

In this study, Mialon and colleagues focus on specific neurons in the worm nerve cord, and find that the IgLON homologs RIG-5 and ZIG-8 localize to pre- and postsynapses, respectively, and trans-neuronally interact (via Ig1) to control the localization of one another. Moreover, ZIG-8 directly binds and recruits ACR-16 via its Ig2 domain. Thus, they uncover a novel mechanism controlling the synaptic localization of an AChR via extracellular interactions with IgSF synaptic molecules.

This is a very elegant study, well-constructed and addressing the majority of the critical aspects required for the authors to establish their hypothesis. It is carefully designed and highly convincing, and its potential implications on other neuronal systems and higher organisms - given the conserved nature of all molecules involved - is of a broad interest to the molecular neuroscience community. I actually think that if the authors would have written the paper in a bit more broader manner, and perhaps focusing on synaptic targeting rather than NTR localization, this study could be published in the top journals. That said - in its current state and with minor textual changes and improvements of figure presentation - this is a top paper for Nature Comm.

1) Annotation of the figures in many places is lacking and they are sometimes confusing or difficult to interpret (in some cases even after reading the entire legend). The system is not straightforward and involves many types of neurons that is difficult for the non-worm expert to follow. There are some schemes throughout which try to clarify, but they are not clear enough (e.g., 1D doesn't show AVE, 1F is tiny and hard

to understand - zoom in would help). Better annotation of the images themselves would also be helpful (e.g. S Fig. 5, and other places - two types of arrows indicate two different neurons - but the legend doesn't say which type points to what).

We thank the reviewer for this advice. We have modified the figures and legends accordingly. Here are the main revisions:

- We have used a consistent code for showing neuromuscular junctions (arrows) and neuron-neuron synapses (arrowheads) throughout the figures, including **Supplementary Figure 5**.

- In **Figure 1d**, we have added a schematic illustrating NeuroSIL expression in DB neurons. We did not label the AVA presynaptic neuron "AVE" because AVA neurons receive inputs from various presynaptic neurons, including but not limited to AVE neurons. Instead, we have modified Figure 1g color display to highlight that AVE is presynaptic to AVA, along with other neurons.

- In **Figure 1f**, we have enlarged schematics of *C. elegans* for improved clarity.

- In **Figure 4**, we have revised the presentation of genetic backgrounds and corresponding synaptic patterns.

- We have added details on the *rig-5* and *zig-8* isoforms used and the deletions made in the main text (line 333), in the Methods (line 873), and in the legend of Supplementary Figure 2. In addition, we have provided the protein sequences of *rig-5* and *zig-8* in a new **Supplementary Table S7**.

Main text: "In contrast, fragments of RIG-5 showing deletions in the Ig2, Ig3, or both Ig2 and Ig3 domains, were detected at synapses and effectively promoted ACR-16-AID-Scarlet concentration in the presence of full-length ZIG-8 (**Supplementary Figure 2 and 6a-c**).

Methods: "The protein sequences of *rig-5* and *zig-8* are provided in **Supplementary Table S7**. The cDNAs were cloned by RT-PCR from mixed-stage *C. elegans* RNAs."

"**Supplementary Figure 2 (related to Figure 2). The *rig-5* and *zig-8* loci.** (A) Genomic organization of *rig-5* and the positions of different tags and mutants used in this study. We used the longest isoform (C36F7.4e.1), considering the second methionine in the transcript as a translation start site, as it generates a strongly predicted signal peptide (see Supplementary Table S7 for sequence numbering and reference³⁰). (B) *zig-8* locus, tags and mutants, including an insertion used in this study. Genome annotation of *C. elegans* reports a single transcript for *zig-8* (Y39E4B.8.1), which encodes a protein with UniProt accession number G5ED00 (Supplementary Table S7). Boxes represent exons with sequences of the Ig domains colored in orange. Ig domains were identified using the SMART web-based tool (Simple Modular Architecture Research Tool, <http://SMART.embl-heidelberg.de>). Complete coding sequence were deleted for the *rig-5* Ig1 and Ig3 domains (*rig-5(bab526)* and *rig-5(bab520)* alleles). In contrast, only partial deletions were created for the *rig-5* Ig2 and *zig-8* Ig1 domains due to the presence of large introns (*rig-5(bab518)* and *zig-8(bab528)* alleles). Additionally, a small portion of the coding sequence in the *zig-8* Ig2 domain was retained because of CRISPR design constraints (*zig-8(kr727)* allele). Arrowheads: point mutations; brackets: deletions. Supplementary Table S2 contains a brief description of *rig-5* and *zig-8* alleles, including deletion boundaries."

Finally, we adjusted the abstract length, subheading, and figure display to align with the journal's style guidelines.

2) The intro is, in my opinion, a bit tedious. Too many facts and gene names. It's a stylish preference - so completely up to the authors - but one could make the paper lighter and more interesting to the general reader.

The most innovative finding of our study is the unexpected binding of a postsynaptic receptor via its extracellular domain. In this context, we found it particularly interesting to precisely compile known instances of such extracellular interactions - a rare mechanism involving only a small number of synaptic adhesion molecules. While we acknowledge that this introduction may seem tedious, we believe our work stands out meaningfully within this body of literature.

3) Tissue specific manipulation - again worm specific, but I could not find specifics on how the authors control the tissue-specificity of the AID experiments (Fig. 1B) - how is the auxin administered/introduced to the animal?

In response to this comment, we have revised the relevant sentence in the Methods section (line 893). The updated wording is highlighted in brown below:

“Auxin-induced cell specific degradation

For all images except in Fig. 4e, f, Fig. 6, Supp. Fig.1b, Supp. Fig. 3a, Supp. Fig. 4 and Supp. Fig. 5, worms were grown on auxin plates to degrade ACR-16-AID-Scarlet at NMJs. **To induce protein degradation**, adult *C. elegans* were transferred to auxin plates, and their progeny was analyzed. Auxin plates were prepared by adding auxin (indole-3-acetic acid, Sigma-Aldrich) from a 400 mM stock solution in ethanol to NGM, at the final concentration of 1 mM²².”

4) The screen - no hard data or information is provided about the genetic screen in which zig8 and rig5 were identified.

As noted in our response to Reviewer 1 point 1 (minor comments), details on the genetic screen have now been included in the Methods section.

5) "dimmer" puncta in 3E - not evident in the fluorescent images.

The dimmer appearance of GFP-RIG-5 puncta in the *acr-16(ok789)* mutant is supported by a reduced fluorescence intensity per puncta, quantified from a set of 15–16 images and shown in the right panel of **Figure 3e**. This difference is also visually apparent: when using the ‘fire’ lookup table, the center of GFP-RIG-5 puncta appears white in the wild type, but shifts to yellow in the *acr-16(ok789)* mutant, indicating lower intensity. To enhance visual clarity for the reader, we have now included the ‘fire’ lookup table directly in **Figure 3** and other **Figures**.

6) "Taken together, our data strongly suggest that the main function of RIG-5 and ZIG-8 is to cluster the nicotinic receptor ACR-16 at specific cholinergic synapses." - I find this a bit too strong of a statement.

We have toned down this statement (line 268):

" Taken together, our data suggest that RIG-5 and ZIG-8 control the clustering of ACR-16 nicotinic receptors at specific cholinergic synapses."

7) In figure 4 - what does the change in the number of puncta mean? this is not really evident in the fluorescent images BTW.

We thank the reviewer for this insightful comment. The ambiguity surrounding the rescue experiments was also noted by Reviewer #1 (Point 2). Please refer to our detailed response and the corresponding revisions in the main text, and **Figure 4**, we have made.

Also - how does the number of puncta relate to the number of synapses (in figure 1) - this is unclear.

We thank the reviewer for the opportunity to clarify this point. First, in the rescue experiments, the number and distribution of puncta closely resembled the pattern typically observed at synapses between AVEs and AVAs, or at synapses formed onto AVAs (**Figure 4a-c**). We have now added this clarification to the manuscript (line 283):

“In these rescue experiments, we observed the recovery of distinct classes of synapses depending on the genetic background.”

Second, we directly demonstrate recovery of AVE>AVA synapses using an AVE-specific presynaptic marker (**Figure 4e**), as noted on line 299. We have modified a sentence to clarify the conclusion of this experiment (in brown):

“To confirm the synaptic location of these puncta, we expressed BFP-RIG-5 in the AVEs along with an AVE-specific presynaptic marker, and Scarlet-ZIG-8 in the AVA, in the double *rig-5(0); zig-8(0)* mutant. We observed that BFP-RIG-5 and Scarlet-ZIG-8 formed puncta juxtaposed to the AVE presynaptic marker, supporting the formation of functional synapses (**Figure 4e**).”

ALSO - if Zig8 and Rig5 interact independently of Acr16 then why does it matter which of the proteins is in the pre- or post- synapse? This relates to 4E/F

When RIG-5 is expressed in the presynaptic neuron and ZIG-8 in the postsynaptic neuron, their interaction is detected. In contrast, reversing the expression pattern does not yield a stable interaction. One possibility is that the RIG-5—ZIG-8 interaction can be stabilized at synapses upon interaction with other synaptic proteins. Interestingly, RIG-5 is a likely candidate for engaging presynaptic components. However, this remains speculative, and we prefer not to extend the interpretation further at this stage.

8) colP does not equal with physical interaction but rather presence in the same complex.

We agree. Due to technical limitations, we have decided to remove this specific experiment from the revised manuscript. Please refer to our response to the reviewer #1's first point.

9) are these worms "normal"?

The behavioral consequences of the *acr-16*, *rig-5*, and *zig-8* mutations remain an open question. We have not observed any obvious defects in forward or backward movement on agar plates. Noteworthy, complete loss of ACR-16 does not cause obvious locomotory defects (PMID: 15917232). Because *rig-5* and *zig-8* mutations impact the connectivity of the locomotory circuit, we cannot rule out that subtle modifications of locomotion might be detectable by detailed quantitative analysis and calcium imaging of neuronal activity. While this would be interesting, we believe it is beyond the scope of our current work.

I want to reiterate - this is an excellent and very elegant paper. One of its main discoveries is the way by which igSF proteins 'match' pre and post synaptic partners - which is by clustering neurotransmitter receptors. Super cool, super relevant - definitely accept.

RESPONSE TO REVIEWERS' COMMENTS

Reviewer #1 (Remarks to the Author):

I appreciate the extensive effort the authors have put in to further demonstrate the interaction between ZIG-8 and ACR16 and agree with their decision to remove the western blot data from the manuscript - the interaction is convincingly supported by other lines of evidence. The added clarifications in response to the reviewers' comments have further improved an already excellent and exciting study - this manuscript is ready for publication.

Response:

We thank the reviewer for taking the time to review our manuscript, and for her/his insightful comments.

Reviewer #2 (Remarks to the Author):

In this revised version of the manuscript the authors addressed most of my comments of what I already thought was a very good manuscript. I do have only one suggestion - the title of the last section of the results "ZIG-8 Ig2 domain binds the ACR-16 extracellular domain" is not, in my opinion, totally supported by the data. Is it the most plausible? likely yes. But I don't think - especially now without the colP (and even with it) that the authors can truly make this claim. If you want to maintain this claim then perform SPR experiments on isolated proteins. This is not necessary in my opinion but then tweak down this statement.

Response:

We agree that a biophysical assay using purified proteins would provide definitive proof of a direct interaction between ACR-16 and ZIG-8. However, we appreciate that the reviewer concurs that a direct interaction remains the most plausible explanation for our findings. We have modified the title of the last section of the results as suggested by the reviewer.

« *The Ig2 domain of ZIG-8 recruits ACR-16* »